_Resource_

# Proteomic aging signatures across mouse organs and life stages

Enzo Scifo [1,3], Sarah Morsy [1,3], Ting Liu [1,3], Kan Xie[1], Kristina Schaaf[1], Daniele Bano [2] & Dan Ehninger [1✉]

## Abstract

**Aging is associated with the accumulation of molecular damage, functional decline, increasing disease prevalence, and ultimately mortality. Although our system-wide understanding of aging has significantly progressed at the genomic and transcriptomic levels, the availability of large-scale proteomic datasets remains limited. To address this gap, we have conducted an unbiased quantitative proteomic analysis in male C57BL/6J mice, examining eight key organs (brain, heart, lung, liver, kidney, spleen, skeletal muscle, and testis) across six life stages (3, 5, 8, 14, 20, and 26-month-old animals). Our results reveal age-associated organ-specific as well as systemic proteomic alterations, with the earliest and most extensive changes observed in the kidney and spleen, followed by liver and lung, while the proteomic profiles of brain, heart, testis, and skeletal muscle remain more stable. Isolation of the non-blood-associated proteome allowed us to identify organ-specific aging processes, including oxidative phosphorylation in the kidney and lipid metabolism in the liver, alongside shared aging signatures. Trajectory and network analyses further reveal key protein hubs linked to age-related proteomic shifts. These results provide a system-level resource of protein changes during aging in mice, and identify potential molecular regulators of age-related decline.**

**Keywords** Aging; Mass Spectrometry; Mouse Organs; Protein Trajectories; SureQuant
**Subject Categories** Methods & Resources; Proteomics

## Introduction

Normal aging is marked by progressive accumulation of molecular damage at the cellular, tissue, and organ levels, leading to functional decline, increased disease prevalence, and ultimately, mortality (Gladyshev et al, 2021). Despite the many theories proposed to explain the biological basis of aging, the precise underlying cellular and molecular mechanisms are still poorly understood (Keshavarz

et al, 2023a, 2023b; Lopez-Otin et al, 2023). Unraveling these mechanisms is critical for the development of therapies aimed at mitigating age-related degeneration and improving healthspan.

Extensive research has documented aging-related changes in the genome, epigenome, and transcriptome (Davie et al, 2018; Sen et al, 2016; Zou et al, 2000). However, it's impact on the proteome, remains less understood. Given that proteins are the primary effectors of cellular function, deciphering proteomic changes during aging is essential for understanding the biological basis of age-related decline. Mass spectrometry-based global proteomics allows researchers to identify and quantify proteins associated with specific cellular processes in complex biological samples, providing insights into cellular function and dynamics (Dengjel et al, 2012; Geyer et al, 2016; Itzhak et al, 2017). However, this approach requires significant instrument time to address stochastic peptide sampling limitations (Bekker-Jensen et al, 2017). On the contrary, targeted mass spectrometry is well suited for quantitation of selected protein targets across various samples and is amenable to relatively low sample amounts (Song et al, 2017; Stopfer et al, 2021).

A landmark human plasma proteomic study involving over 4000 healthy adults revealed that aging is a dynamic, non-linear process, marked by waves of protein changes (Lehallier et al, 2019). However, comparable comprehensive analyses across multiple organs in mammalian models remain scarce. Recent studies profiling proteomic changes across 8–10 tissues from mice identified significant age-associated changes, particularly in mitochondrial proteins, immune-related factors, and macro complex compositions (Keele et al, 2023; Takasugi et al, 2024). For instance, age-related dysregulation of the mitochondrial interactome in 30-month-old skeletal muscle showed significant changes in mitochondrial respiratory complexes I and IV, in addition to enzymes involved in fatty acid oxidation and TCA cycle (Bakhtina et al, 2023). Similarly, chromatin proteome analysis across six organs in mice (3–15 months) reported gradual protein changes in the brain, heart, and kidney, while changes in the lung and liver were more pronounced between 5 and 10 months. In contrast, the spleen showed relatively few alterations over the same period, with each organ displaying distinct proteomic signatures (Oliviero et al, 2022). Moreover, proteomic studies in rodents have often reported minimal age-associated protein changes (Angelidis et al, 2019;

[1]Translational Biogerontology Lab, German Center for Neurodegenerative Diseases (DZNE), Venusberg-Campus 1/99, 53127 Bonn, Germany. [2]Aging and Neurodegeneration Lab, German Center for Neurodegenerative Diseases (DZNE), Venusberg-Campus 1/99, 53127 Bonn, Germany. [3]These authors contributed equally: Enzo Scifo, Sarah Morsy, Ting Liu. ✉E-mail: Dan.Ehninger@dzne.de

Oliviero et al, 2022; Ori et al, 2015; Walther and Mann, 2011; Yu et al, 2020), though these findings should be interpreted cautiously given the methodological limitations.

The above-mentioned studies, focused on single organs, either had limited age ranges or performed pairwise comparisons, providing an incomplete picture of systemic aging. Moreover, small sample sizes have constrained statistical power, while comparisons between only two age groups—a common approach in aging research—may oversimplify the continuous nature of biological aging and lead to biased interpretations. In addition, while trajectory analyses have been employed to assess age-related protein expression patterns (Lehallier et al, 2019; Coenen et al, 2023), there has been limited exploration of the regulatory architecture underlying these changes, leaving open questions regarding the key molecular players that drive proteomic alterations across different tissues.

To address these challenges, we conducted a large-scale, multi-organ proteomic analysis across the adult lifespan of male C57BL/6J mice. We chose to carry out the present study in C57BL/6J mice, as this work is part of a larger research program investigating the effects of single-gene mutations on aging-associated proteomic changes. These mutations are maintained on a C57BL/6J genetic background, and prior studies have demonstrated lifespan extension in this strain. Moreover, C57BL/6 mice were also recently employed in recent large-scale multi-organ aging studies (Schaum et al, 2020; Takasugi et al, 2024), thus allowing for comparative analysis with our study. By examining eight organs—brain, heart, lung, liver, kidney, spleen, skeletal muscle, and testis—across six time points (3, 5, 8, 14, 20, and 26 months) with a relatively large sample size ($n = 45$), our study provides a detailed temporal landscape of protein expression changes during aging. Importantly, we employed a moderated F-test, a statistically robust alternative to pairwise comparisons, enabling the detection of gradual and systemic proteomic alterations rather than changes confined to isolated time points. This advantage became evident when initial pairwise comparisons between 20-month and 3-month reference samples—an approach widely used in aging studies—yielded fewer differentially expressed proteins than the moderated F-test. Although F-tests have not been widely used in proteomics, some recent studies (Myers et al, 2019; Sebastiani et al, 2021) successfully employed them. The use of the moderated F-test allows for more powerful and stable inference to detect significant changes in protein abundance compared to ordinary $t$ tests (Kammers et al, 2015). In addition, we integrated trajectory analysis, Gene Ontology (GO) enrichment, and network-based protein hub identification to uncover both shared and organ-specific molecular changes associated with aging.

Our findings reveal that proteomic alterations are not uniformly distributed across organs but instead exhibit distinct temporal and tissue-specific patterns. Notably, we observed the most pronounced proteomic changes at 20 months, a time point preceding detectable mortality in male B6 mice (Xie et al, 2022). The kidney and spleen displayed the highest number of differentially expressed proteins (DEPs), followed by the liver and lung, while the brain, heart, testis, and skeletal muscle exhibited more stable proteomes. Network analysis identified key regulatory hubs that may drive age-associated proteomic remodeling, providing novel insights into the molecular coordination of aging across organ systems. Functional enrichment analyses further highlighted both shared

and organ-specific pathways, with oxidative phosphorylation in the kidney, cytoplasmic translation in the spleen, lipid metabolism in the liver, and extracellular matrix organization in the lung emerging as central aging-associated processes.

To further validate our findings, we employed SureQuant-based quantitative targeted mass spectrometry on a subset of differentially expressed proteins identified from the global proteomics experiment. We confirmed several age-associated proteins that were differentially expressed in one or more of the four organs—kidney, spleen, liver, and lung—which showed the most pronounced changes at 20 months. Notably, many of these validated proteins were shared across multiple organs. Identified peptides were confirmed by SureQuant-based MS analysis with validation rates of 80.40%, 76.10%, 78.10%, and 54.8% in the kidney, spleen, liver, and lung, respectively.

By integrating large-scale proteomics with advanced statistical and network-based approaches, our study offers a more comprehensive view of age-related protein expression dynamics than previous investigations. The identification of key protein hubs across multiple tissues not only enhances our understanding of systemic aging but also provides potential targets for future studies aimed at mitigating age-related functional decline. These findings underscore the importance of holistic, multi-organ approaches in aging research and set the stage for further exploration into the molecular mechanisms that drive aging at the protein level.

## Results

### Optimal design to detect fine-grained age-dependent changes in the mouse proteome

To investigate age-associated protein changes, we isolated eight organs (brain, heart, lung, liver, kidney, spleen, skeletal muscle, and testis) from male C57BL/6J mice across six time points for global and targeted proteomics (Fig. 1A–D). These time points covered four life stages: young adults (3 and 5 months), adult (8 months), midlife (14 months), and late life (20 and 26 months). Our experimental design (Fig. 1A) included a relatively large sample size ($n = 45$), with at least five animals per time point, enhancing the reliability of our findings.

In total, we identified 8814 proteins across the eight mouse tissues examined, with each organ yielding at least 4000 proteins, except for heart and skeletal muscle (Dataset EV1). Protein extraction from these two organs was likely hindered by their muscular tissue composition. Dimensionality reduction of the mouse organs using uniform manifold approximation and projection (UMAP) indicated a clear separation of the various organs (Fig. 1B). We confirmed the reproducibility of our MS datasets based on the comparable number of identified total proteins at different time points within the same organ.

### Substantial differences in the extent to which aging affects proteomic changes in various mouse tissues

To assess overall age-associated protein changes across the various time points, we employed a moderated F-test (Singer and Hughey, 2019; Smyth, 2004) to identify differentially expressed proteins (DEPs) across the eight mouse organs. Our analysis revealed that

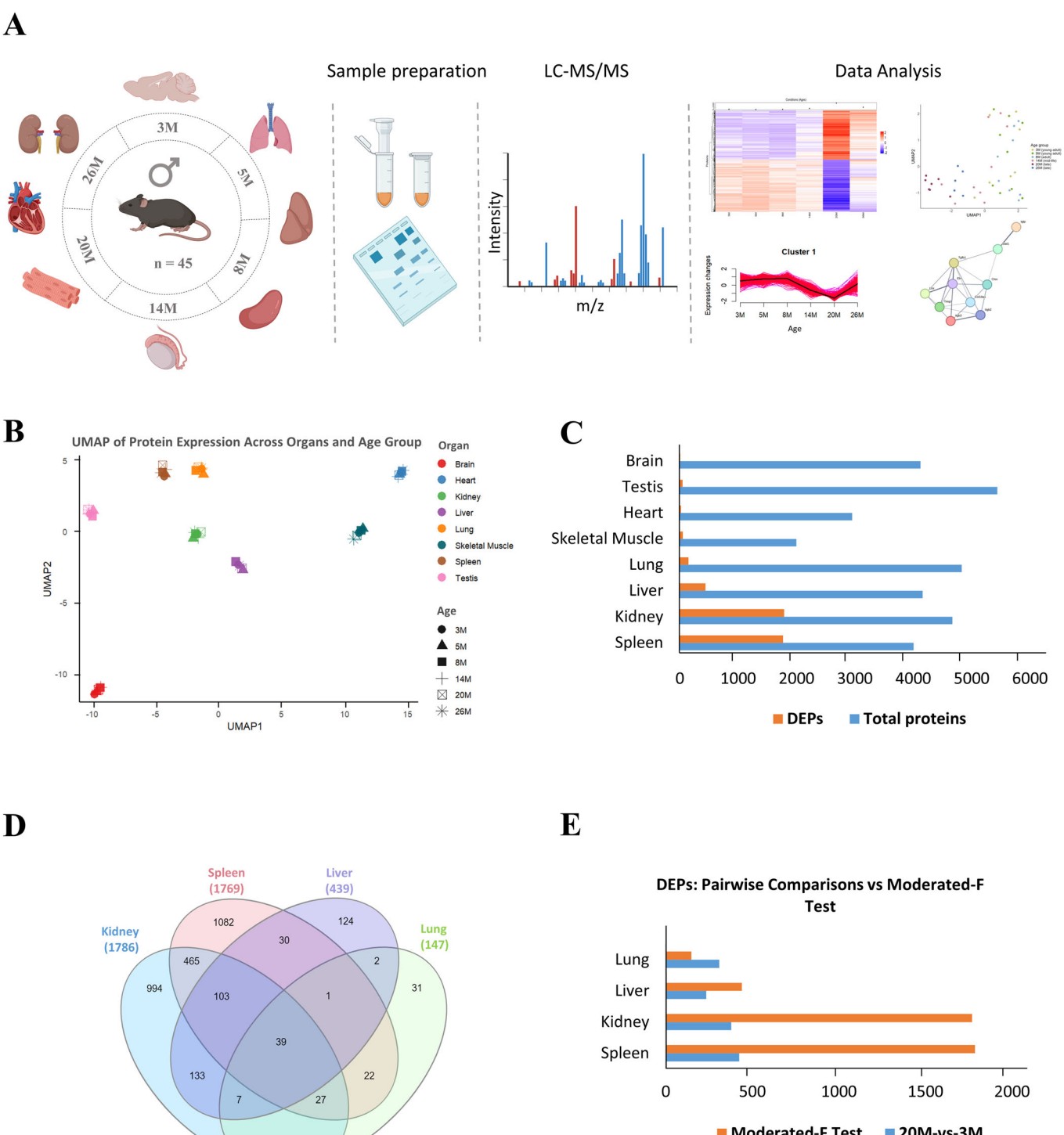

the proportion and absolute count of age-associated differentially expressed proteins (DEPs) varied significantly across the assessed mouse organs. The highest numbers and percentages of DEPs were observed in the spleen (1769 DEPs, 44.2%) and kidney (1786 DEPs, 38.3%), followed by the liver (439 DEPs, 10.6%) and lung (147 DEPs, 3%). In contrast, skeletal muscle (52 DEPs, 2.6%), testis (49 DEPs, 0.9%), heart (18 DEPs, 0.6%), and brain (4 DEPs, 0.1%)

showed markedly lower counts and percentages of DEPs (Fig. 1C; Dataset EV2).

Moreover, the number of identified DEPs was not biased by the total number of proteins since the testis, brain and lung with relatively high numbers of total proteins still had very few DEPs. Kidney and spleen, kidney and liver, as well as kidney, spleen and liver had the most shared DEPs (Fig. 1D).

**Figure 1. Experimental design, workflow and distribution of differentially expressed proteins (DEPs) across organs.**

(A) Schematic representation of the experimental workflow. Male C57BL/6J mice at six different age points (3, 5, 8, 14, 20, and 26 months) were used to examine age-related protein expression changes across multiple organs (brain, heart, kidney, liver, lung, skeletal muscle, spleen and testis) from a single experiment. M—denotes age group in months. The following number of biological replicates were used in the study for each age group: 7 (3 and 20 months), 9 (5 and 14 months), 8 (8 months) and 5 (26 months). The workflow includes tissue collection, protein extraction, and mass spectrometry analysis. Data processing steps included protein quantification, clustering of DEPs, trajectory analysis, and network analysis for the identification of age-associated protein hubs. (B) UMAP plot of shared total identified proteins across organs. Individual organs and age groups are represented by different colors and shapes, respectively. (C) Summary of the total identified proteins and DEPs for each organ. The total number of identified proteins and significantly differentially expressed proteins for each organ, across age groups, are represented by blue and orange bars, respectively. (D) Venn diagram indicating overlap of DEPs in the kidney, liver, lung, and spleen. (E) Contrast of Pairwise comparisons (20-month vs 3-month reference) versus moderated F-test for the kidney, liver, lung and spleen. https://app.biorender.com/illustrations/677d53bf42d2236f53575092 https://app.biorender.com/illustrations/6724f44102db9f70a8ecea5b. Source data are available online for this figure.

In contrast to the pairwise comparisons which inherently only capture changes across a given set of two time points, the moderated—F-test allow for a holistic view of changes across time and minimize potential biases from any given two time points. Unlike pairwise comparisons, which assess differences between only two time points at a time, the moderated F-test provides a global statistical framework that evaluates expression changes across all time points simultaneously, reducing biases that arise from isolated pairwise analyses. To illustrate this, we performed pairwise comparisons between 20-month-old mice and 3-month-old reference controls—an approach commonly used in proteomic aging studies (Takasugi et al, 2024). The percentages of DEPs identified using pairwise comparisons were relatively low (kidney: 7.3%, liver: 5.1%, lung: 5.7%, spleen: 9.6%) (Dataset EV1) and closely aligned with those reported in previous large-scale studies. However, applying the moderated F-test substantially increased the number of identified DEPs (Fig. 1E), providing a more comprehensive and statistically robust view of aging-associated proteomic shifts. Furthermore, we evaluated the performance of the moderated F-test in comparison to established statistical tests-the standard F-test and linear trend analysis, for identification of DEPs in the four organs with most protein changes. This analysis indicated highly concordant results between the moderated and classical F-tests with a similar number of identified DEPs, i.e., ≥94% overlap (Dataset EV2). The moderated F-test yielded slightly more DEPs for the kidney and spleen in comparison to the linear trend analysis. Although we identified more DEPs for the liver and lung with the linear trend in comparison to the moderated F-test, the extra identifications from the linear trend had marginal $R^2$ ≤0.35 (Dataset EV2). As such, they may represent either false positives or low-confidence DEPs. Importantly, there is a substantial overlap of DEPs identified by the moderated F-test with those from the linear trend, i.e., 86%, 73%, 88% and 96% for the kidney, spleen, liver and lung (Dataset EV2). We therefore opted for the more conservative statistical approach (moderated F-test) that yielded robust findings, for this study.

## Protein expression changes were most dynamic between 14 and 20 months across several tissues

To investigate the temporal dynamics of protein expression during aging, we evaluated the relative abundances of differentially expressed proteins (DEPs) across various mouse organs and age groups. This analysis aims to identify specific ages associated with peak proteomic alterations and examine how aging differentially affects individual organs. Heatmap representations of protein expression patterns for DEPs associated with individual organs indicated that most protein changes occurred between 14 and 20 months (Figs. 2A–D and EV1). Given the relatively few DEPs obtained in skeletal muscle, testis, heart, and brain, we focused our cluster analysis on the four primary organs with the most protein changes: kidney, spleen, liver, and lung. This analysis revealed distinct patterns for the age-associated protein changes within these organs. Most identified DEPs were downregulated across many age groups (Fig. 2A–D). In the kidney, protein expression remained relatively stable between 3 and 8 months with changes emerging between 8 and 14 months and becoming most pronounced in 20 months old animals (Fig. 2A). In the spleen, stable protein expression was observed between 3 and 8 months, followed by significant changes occurring between 8 and 20 months, which then diminished by 26 months (Fig. 2B). Conversely, the liver showed relatively stable protein expression from 3 to 14 months, with distinct changes emerging between 14 and 20 months (Fig. 2C). The protein expression patterns in the lung indicated stable levels from 3 to 8 months, with initial changes becoming evident between 8 and 14 months, progressing significantly from 14 to 20 months, and slightly diminishing by 26 months (Fig. 2D). These patterns suggest that aging affects each organ differently, with kidney and spleen exhibiting more dynamic shifts compared to liver and lung. The line graphs accompanying the heatmaps further illustrate the trajectory of these changes, while the associated PC1 scores capture part of the variance (79.1–90.2%) (Dataset EV3) in protein expression across ages and hence reflect the complex and organ-specific nature of age-related proteomic alterations.

Based on the protein expression patterns displayed in the heatmaps, we determined peak protein changes in most assessed mouse organs at 20 months. Dimensionality reduction using uniform manifold approximation and projection (UMAP) (Becht et al, 2019), also indicated separation of the 20-month age group from the others (Fig. 2E–H). Moreover, we observed heterogeneity in the differential protein expression across the various mouse organs. For instance, the kidney and spleen exhibited a strong age effect as evidenced by the high number of protein changes, while a moderate age effect was observed in the liver and lung. The other 4 assessed organs showed minimal protein changes (Figs. 1B and EV1). This organ-specific variability in age-dependent protein expression changes underscores the complexity of aging at the molecular level and suggests that certain organs may uniquely change during aging, potentially reflecting differing functional demands or resilience mechanisms.

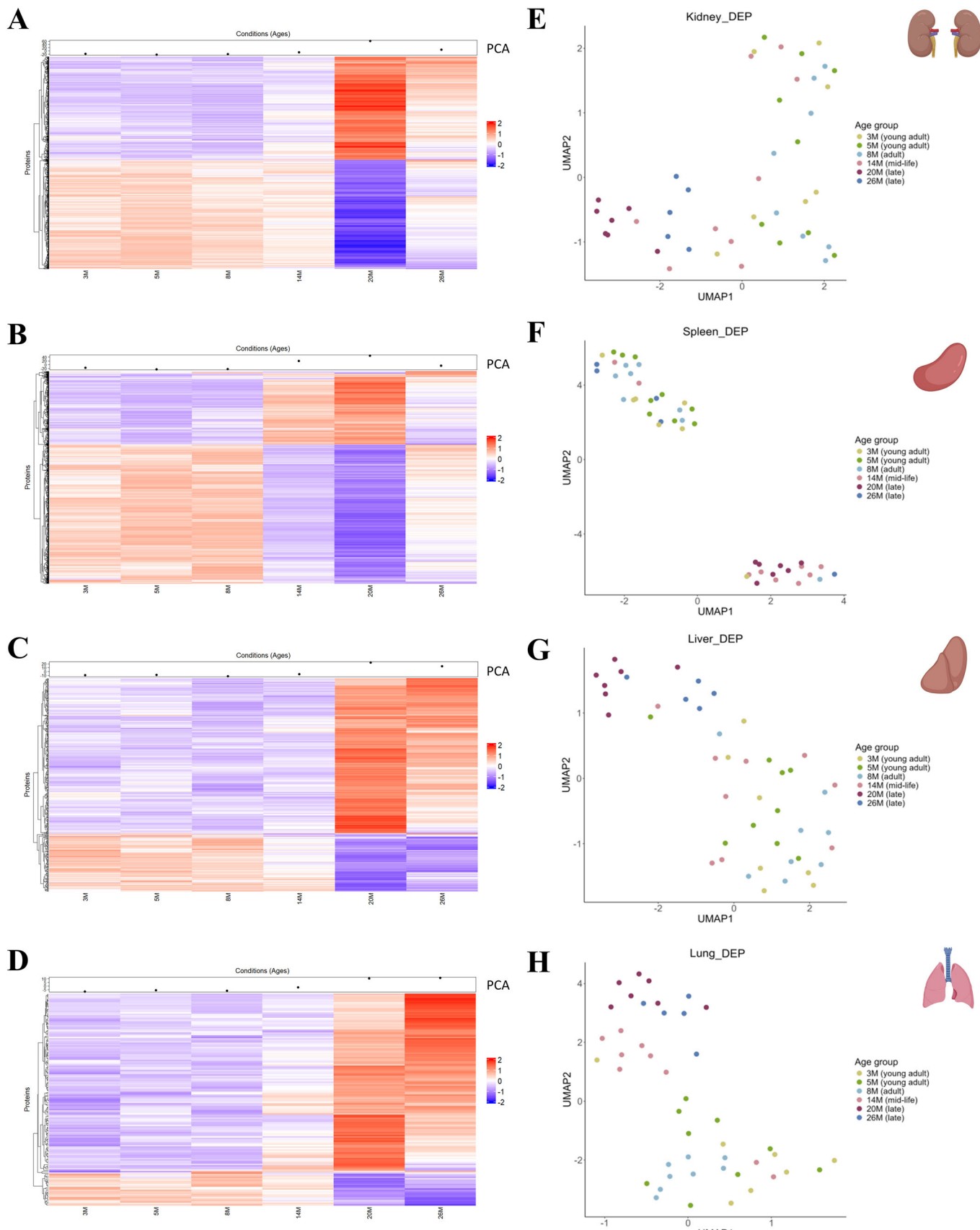

**Figure 2.  Differential expression profiles and UMAP analysis of age-associated protein changes in kidney, spleen, liver, and lung.**

(A–D) Heatmaps of differentially expressed proteins (DEPs) across six age groups (3, 5, 8, 14, 20, and 26 months) for each organ: kidney (A), spleen (B), liver (C), and lung (D). Proteins (rows) are clustered based on expression patterns across ages (columns). The color gradient represents log-transformed expression levels, with red indicating upregulation and blue indicating downregulation relative to each protein's mean expression. Adjacent line graphs depict the trajectories of mean z-scored expression levels, with age on the x axis and expression changes on the y axis. The top dot plots display the principal component 1 (PC1) trajectory of the z-scored expression profiles, highlighting shifts in protein expression across ages. Variance explained by PC1 is 90%, 87.4%, 84%, and 79% for the kidney, spleen, liver, and lung, respectively. (E–H) UMAP plots of DEPs for each organ: kidney (E), spleen (F), liver (G), and lung (H). Each dot represents an age group, color-coded by age. Clusters in UMAP space illustrate age-related divergence in protein expression patterns, with distinct trajectories observed for each organ over time. Differential expression and UMAP analyses are based on 5–9 biological replicates for the various age groups: 7 (3 and 20 months), 9 (5 and 14 months), 8 (8 months), and 5 (26 months). Source data are available on the Data Dryad public repository for this figure.

## Temporal protein expression analysis identified the most enriched age-associated clusters

Next, we aimed to explore the dynamic progression of age-associated protein changes across organs and therefore employed fuzzy c-means (FCM) clustering to identify temporal patterns within DEPs at various age groups (Kumar and E Futschik, 2007). Based on partition coefficient (PC) and partition entropy (PE) metrics, we determined the optimal cluster numbers for each organ: three clusters for kidney and lung, two for the spleen, and four for liver. We did not perform any clustering for skeletal muscle, testis, heart and brain, for which we had very few differentially expressed proteins (Figs. 1C and 3A–D). Furthermore, we selected the most enriched clusters for each organ according to high mean Membership and low SD Membership statistics. For the kidney, cluster 3, associated with small molecule metabolic processes, was the most enriched. This cluster comprised DEPs that exhibited relatively stable expression between 3 and 14 months, followed by a steep decrease from 14 to 20 months, and a rapid increase from 20 to 26 months (Figs. 3A and EV2; Dataset EV4). In the spleen, cluster 1, linked to cellular metabolic processes, was the most enriched, although cluster 2, associated with wound healing, was also notable (Figs. 3B and EV2). The DEPs in cluster 1 showed stable expression between 3 and 8 months, a progressive decrease from 8 to 20 months, and an increase from 20 to 26 months. Cluster 2 exhibited similar stable expressions between 3 and 8 months, followed by a progressive increase between 8 and 20 months and a decline at 26 months (Figs. 3B and EV2; Dataset EV4). In the liver, cluster 4 (small molecule metabolic process) was most prominent, characterized by DEPs that maintained stable expression between 3 and 8 months, decreased slightly between 8 and 14 months, followed by a stronger decrease from 14 to 20 months and a partial recovery at 26 months (Figs. 3C and EV2; Dataset EV4). Meanwhile, cluster 1 (negative regulation of hydrolytic activity) in the lung indicated stable protein expression between 3 and 8 months, a slight increase between 8 and 14 months, an accelerated rise from 14 to 20 months, and a decline at 26 months (Figs. 3D and EV2; Dataset EV4). The temporal patterns observed in protein expression in both heatmaps, and trajectory analysis of organ clusters highlight distinctive patterns for each organ related to their starting points and magnitude of changes. This becomes evident when evaluating the patterns of eight identified shared proteins across the four organs.

Notably, when evaluating shared differentially expressed proteins across the 4 mouse organs with the most DEPs, the spleen and kidney exhibited more pronounced age-dependent protein changes. In comparison, we observed moderate protein changes in liver and lung (Dataset EV4). These changes typically began between 8 and 14 months, peaked at 20 months, and started to decline by 26 months (Fig. 3E). We first analyzed age-related trajectories of PLD4 and HEXB. Phospholipase D family member 4 (PLD4) regulates cytokine production for inflammatory response (Gavin et al, 2018). Hexosaminidase subunit beta (HEXB), in conjunction with the cofactor GM2 activator protein, catalyzes the degradation of the ganglioside GM2 and other molecules containing terminal N-acetyl hexosamines (Lecommandeur et al, 2017). Both were relatively stable from 3 to 8 months, gradually increased between 8 and 14 months and sharply increased from 14 to 20 months. Protein levels of PLD4 increased at 26 months, except in the kidney, where they decreased. In contrast, HEXB protein expression declined at 26 months, except in the spleen, where it increased. We also assessed the age-related trajectories of ARPC1B, STAT1, and TAPBP. Actin-related protein 2/3 complex, subunit 1B (ARPC1B) is a cytoskeleton protein involved in Arp2/3 complex-mediated actin nucleation (Gournier et al, 2001). Signal transducer and activator of transcription 1 (STAT1) mediates cellular responses to interferons (IFNs), cytokines and growth factors (Nair et al, 2002). TAP binding protein (TAPBP) acts in antigen processing and presentation of exogenous peptide antigen via MHC class I (Teng et al, 2002). ARPC1B, STAT1 and TAPBP protein levels were mostly stable between 3 and 14 months, followed by a gradual increase in lung and liver from 14 to 20 months and either a slight reduction or increase from 20 to 26 months. The expression patterns in the kidney indicated a rapid increase from 14 to 20 months, followed by a steep decline at 26 months. Expression levels of the three proteins were stable in the spleen between 3 and 8 months, declined sharply from 8 to 20 months and increased expression between 20 and 26 months (Fig. 3E). The above-mentioned examples underscore the varying starting points and magnitude of age-associated protein changes across different tissues.

## Limited overlap of DEPs across tissues

To effectively illustrate the distribution and relationships among differentially expressed proteins across mouse organs, we employed a chord diagram and an UpSet plot (Fig. 4A,B), which visually represent the weighted connections and intersections among DEPs across organs. Notably, the organ pairs: kidney–spleen, kidney–liver, and spleen–liver exhibited the largest overlap in DEPs, with these pairs sharing 634, 282, and 173 DEPs, respectively. The kidney, spleen and liver had 142 shared DEPs (Fig. 4A,B). Moreover, we determined that the overlapping DEPs across the above organ pairs were not random based on their

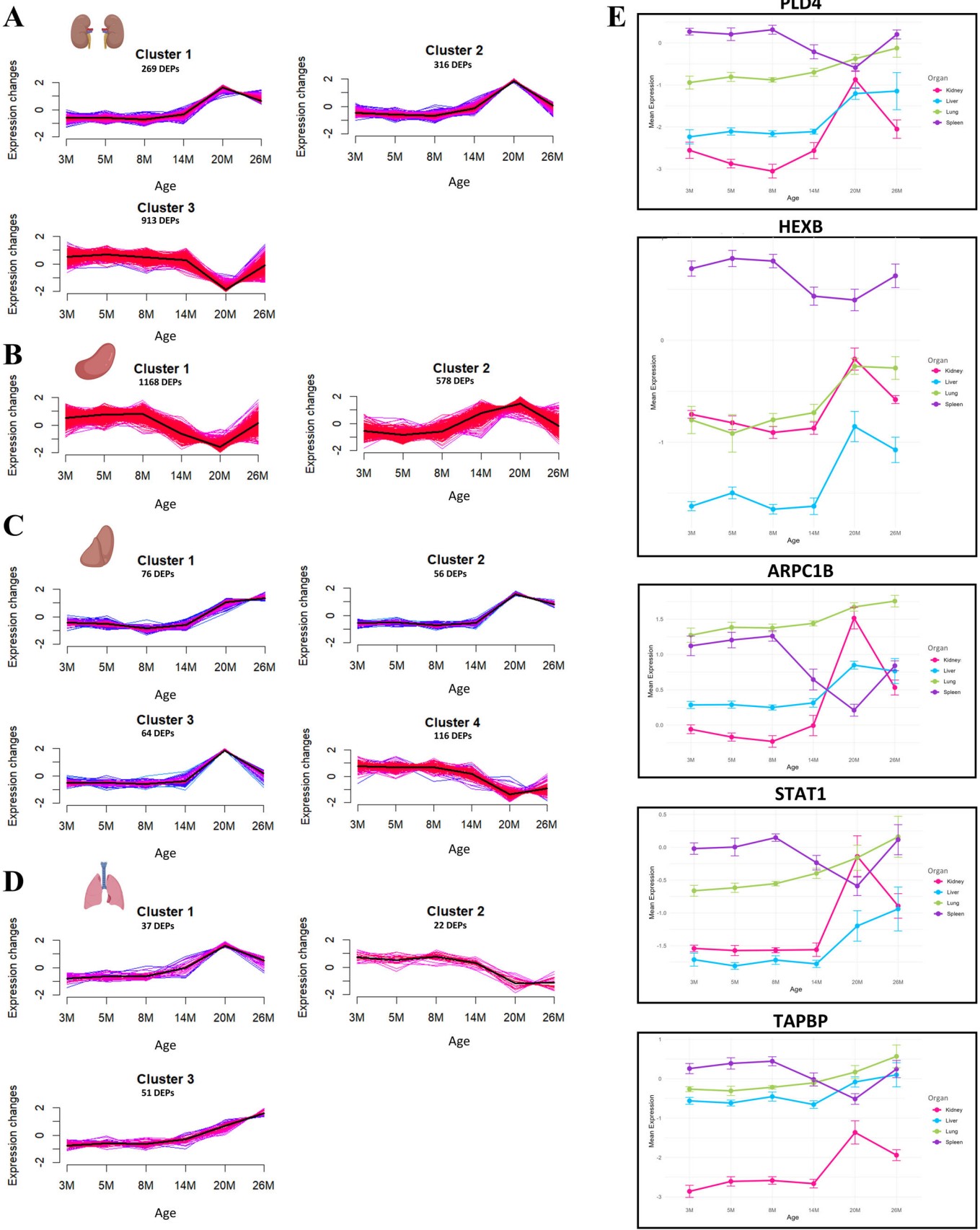

**Figure 3.  Temporal expression dynamics of age-associated differentially expressed proteins of the four organs with most protein changes depicted by trajectory analysis and selected expression profiles.**

(A–D) Mfuzz clustering of z-scored expression levels of differentially expressed proteins (DEPs) showing age-associated expression trajectories in kidney (**A**), spleen (**B**), liver (**C**), and lung (**D**). Each panel displays the distinct temporal expression patterns for multiple clusters identified in each organ, with average trajectories highlighted. Clustering reveals organ-specific patterns in the timing and progression of age-related changes, with shifts in expression occurring at different stages across organs. (**E**) Line graphs illustrating the expression dynamics of selected key age-associated proteins (PLD4, HEXB, ARPC1B, STAT1, and TAPBP) shared between the kidney, spleen, liver and lung over the six age points. Error bars (SEM) of the line graphs for the assessed proteins in a single experiment with 5–9 biological replicates are shown. Each line color represents a different organ, demonstrating organ-specific temporal expression profiles for these proteins. Source data are available on the Data Dryad public repository for this figure.

corresponding hypergeometric probabilities of $2.46 \times 10^{-64}$, $5.25 \times 10^{-95}$ and $1.37 \times 10^{-18}$, respectively. Similarly, the overlapping DEPs across the kidney, spleen and liver were identified with a hypergeometric probability of $2.69 \times 10^{-149}$ (Table EV1). For comparison, 64.2% and 39.4% DEPs in the liver were also identified in the kidney and spleen, respectively. A similar proportion of DEPs in the spleen was identified in the kidney and vice versa (35.8% and 35.5%, respectively). The data further revealed that a substantial proportion of DEPs in certain organs were organ-dominant, with 75% of DEPs in the brain, 60.5% in the spleen, and 55% in the kidney not being shared with other organs (Fig. 4B). In contrast, only 16.7% of DEPs in the heart and 20.4% in the lung were prominent in these organs, indicating a higher level of protein expression overlap in these tissues. Across the various organs, we identified 850 DEPs shared between at least two organs, with 133 proteins common across three or more organs. Interestingly, a subset of DEPs—33, 10, and 9 proteins—were shared across four, five, and six organs, respectively (Fig. 4B; Dataset EV5).

Among these shared proteins, ceruloplasmin (CP) was present in all assessed organs except the brain, while other frequently shared DEPs included immunoglobulins (IGHM and IGKC), haptoglobin (HP), fibrinogens (FGA and FGG), collagen alpha-1(XVIII) chain (COL18A1), complement factor H (CFH), and clusterin (APO-J) (Dataset EV5). Notably, 95 out of the 133 shared DEPs among three organs were found in kidney, spleen, and liver (Fig. 4B). This group included well-known age-associated proteins such as complement factor B (CFB), decorin (PG-S2), syndecan-4 (SYND4), alpha-1-acid glycoprotein 1 (AGP-1), and biglycan (PG-S1), corroborating previous findings (Lofaro et al, 2021). Of note, 22 of the 33 shared DEPs across four organs, were common to the kidney, spleen, liver, and lung, including key proteins like vitronectin (VN), fibronectin (FN), and complement C4-B (C4-B) (Fig. 4B; Dataset EV5). Moreover, three (HPRT1, MYH3 and TTC22) out of eighteen DEPs associated with the aging heart were prominent in that organ (Dataset EV2). In contrast to the overlapping age-associated DEPs observed in the other six organs, the three DEP (HAPLN2, GFAP and PPP3CC) specific to the brain did not show changes in other organs (Dataset EV2), indicating distinct age-related protein expression dynamics in the central nervous system.

Furthermore, GO analysis of shared DEPs across organs revealed dysregulation of critical biological processes associated with aging, such as stress response, defense response, and negative regulation of blood coagulation and fibrinolysis (Fig. 4C). Organ-specific protein network enrichments performed in STRING, highlighted metabolic processes in kidney, small molecule metabolic processes in liver and spleen, and blood coagulation processes in lung (Fig. 4D–G). These findings suggest that while some age-

associated processes have a universal impact across multiple organs, others have a more pronounced effect in individual organs or between a few of them.

## The non-blood tissue proteome revealed organ-specific and shared age-associated protein signatures

Given that we used non-perfused organs in the study, we sought to minimize the contribution of blood to the shared DEPs across organs and therefore dissociated the non-blood-associated proteome from the total proteome for purposes of identifying organ-specific protein signatures.

We initially observed a strong blood-related signal in our GO analysis, both within some organ-specific clusters and shared biological processes across multiple organs (Fig. EV2). This was anticipated, as our study utilized non-perfused mouse organs, leading to the detection of certain proteins linked to the blood proteome. To refine our focus and extract organ-dominant aging signatures, we removed all blood-associated proteins from our differentially expressed proteins (Dataset EV6). By filtering out these blood-derived proteins, we were able to disentangle the effect of blood-derived proteins and obtain a clearer view of organ-dominant dynamics of age-associated protein changes. The list of mouse blood-associated proteins was compiled from a previously published dataset (Niu et al, 2019), supplemented with identifications from own blood proteomics experiments (unpublished data) (Dataset EV6). The resulting differentially expressed non-blood proteome comprised 1388, 1354, 295, and 70 age-DEPs for the kidney, spleen, liver, and lung, respectively. In the remaining 4 mouse organs with fewer age-DEPs, the removal of blood-derived proteins resulted in 27, 22, 8, and 4 organ-dominant DEPs in the skeletal muscle, testis, heart, and brain, respectively (Dataset EV6). Shared non-blood-derived DEPs across the four assessed main organs included: LGALS3BP, ARPC1B, COL18A1, ELN, PLD4, STAT1, HEXB, and TAPBP (Fig. 3E; Dataset EV7).

To identify the most important age-associated biological processes for the four mouse organs with the most DEPs, we input their non-blood-proteome-derived DEPs into Cytoscape and determined the top 100 (kidney and spleen), top 50 (liver) or top 10 (lungs) protein hubs based on maximal clique centrality (MCC) (Dataset EV8). This analysis yielded the most enriched protein networks for the organs depicted in Fig. 5. The kidney protein network consists of two main dense clusters of mitochondrial proteins: the first predominantly comprises mitochondrial ribosomal proteins (top), while the second is enriched in proteins associated with mitochondrial energy production (bottom), including Complexes I, III, IV, and V subunits (Fig. 5A,B). In the spleen, a single dense cluster is primarily composed of ribosomal proteins.

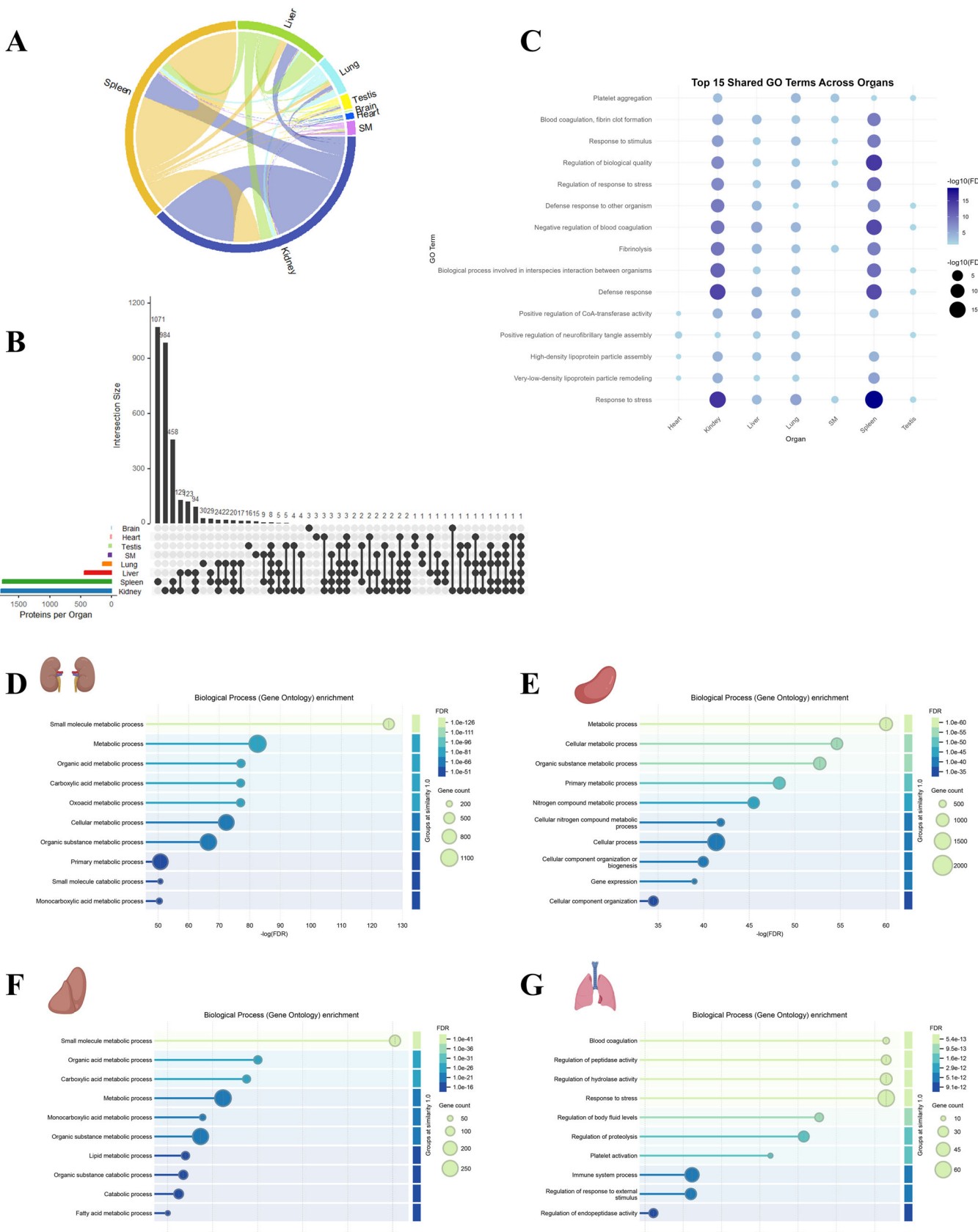

**Figure 4. Age-associated differentially expressed proteins across mouse organs and gene ontology analysis.**

Relationships between overlapping DEPs across mouse organs are depicted in a chord diagram (**A**), illustrating shared and unique DEPs associated with aging in each organ. (**B**) UpSet plot showing intersections of identified DEPs across the assessed eight mouse organs. Bar heights represent the number of unique and shared proteins for individual and organ combinations, respectively. Organs with overlapping DEPs are indicated by dots in the matrix. (**C**) Dot plot displaying the top 15 most enriched GO biological processes across organs. Dot size and color intensity represent the number of associated DEPs and statistical significance (FDR), respectively. Functional annotation of GO biological processes associated with DEPs identified in the kidney (**D**), spleen (**E**), liver (**F**), and lung (**G**). GO enrichment analysis was done using STRING. Bubble size represents the number of associated proteins and the color gradient reflects the significance of enrichment, with lighter and darker shades indicating lower and higher FDR values, respectively. SM skeletal muscle. Source data are available online for this figure.

This cluster showed enrichment in proteins related to RNA Processing and Ribosome Biogenesis, Ribosomal Protein Synthesis and Assembly, Elongation Factors, and Protein Folding and Stress Response (Fig. 5C,D). In contrast, the liver protein network includes three sparse clusters, with the most prominent cluster enriched in fatty acid oxidation and Cytochrome P450 enzymes (Fig. 5E,F). Finally, the protein network of the lung shows a single sparse cluster with an overrepresentation of extracellular matrix (ECM) remodeling-associated proteins (Fig. 5G,H).

Moreover, gene ontology analysis of the protein hubs in STRING indicated the most overrepresented biological processes for the four organs as follows: oxidative phosphorylation (kidney), cytoplasmic translation (spleen), lipid metabolic process (liver) and extracellular matrix organization (lung) (Fig. 5A–H). In addition, we performed similar analysis for organ pairs with the most shared DEPs, namely: kidney and spleen (450), kidney and liver (182), as well as spleen and liver (98). The protein network of shared DEPs between the kidney and spleen further indicated that proteins associated with the mitochondrial electron transport chain/ATP production, as well as oxidative stress and antioxidant defense, were particularly overrepresented in a single cluster (Fig. EV3). The two main clusters arising from the shared DEPs between the kidney and liver showed enrichment in fatty acid and energy metabolism, as well as extracellular matrix (ECM) remodeling-associated proteins. Finally, immune regulation and antigen presentation, as well as extracellular matrix (ECM) remodeling and fibrosis, were the two most overrepresented clusters from the shared DEPs between the spleen and liver (Fig. EV3). The above-mentioned organ pairs indicated overrepresentation in the following biological processes: cellular respiration, fatty acid beta-oxidation, and Arp2/3 complex-mediated actin nucleation, respectively (Fig. EV3).

## Organ-specific, age-related protein dynamics revealed by linear mixed-effects modeling

To uncover tissue-specific patterns of age-related proteomic changes, we applied a linear mixed-effects model (LMM) across combined data from seven murine organs. By including a random intercept per mouse, this model accounted for intra-mouse correlations due to matched tissue sampling, thereby increasing statistical power. Specifically, the model tested for an interaction between age and tissue: Expression ~ Age * Tissue + (1|Mouse). A likelihood ratio test (LRT) between the full and reduced models (excluding the interaction term) identified 2023 proteins with significant age-by-tissue interactions (adjusted $P < 0.05$) (Dataset EV9), indicating divergent age trajectories across tissues.

We further dissected these patterns by performing linear trend tests independently within each organ for each of the significant proteins, estimating both the direction and significance of age-related expression changes. This analysis enabled the construction of a comprehensive matrix summarizing directionality (Up/Down/No Change) across organs for each protein (Dataset EV9).

## Divergent and convergent aging signatures across organs

We leveraged this matrix to categorize proteins into groups based on the consistency or divergence of their aging trajectories across organs. Across kidney, liver, and spleen, consistently upregulated proteins were enriched for immune-related processes, including immune response, cytoskeletal remodeling, and negative regulation of proteolysis, suggesting a shared immune activation signature (Dataset EV9). In contrast, proteins consistently downregulated across these organs showed no clear enrichment in biological processes.

Strikingly, divergent proteins between pairs of organs revealed tissue-specific aging hallmarks. For instance, proteins upregulated in the liver but downregulated in the spleen were enriched for mRNA splicing, RNA processing, and ribosome biogenesis. Proteins upregulated in the kidney and downregulated in the spleen were associated with fatty acid metabolism, oxidation-reduction processes, and actin filament regulation (Dataset EV9), potentially reflecting more robust metabolic and cytoskeletal regulatory activity in the kidney relative to the spleen. Moreover, upregulated proteins in the lung but downregulated in the kidney or liver, consistently mapped to mitochondrial function, TCA cycle, and oxidative phosphorylation (Dataset EV9), implying preserved or even enhanced energy metabolism in the lung with age, contrasting with a decline in classical metabolic organs.

This comparative approach also identified a subset of proteins with organ-specific expression trends (i.e., significantly regulated with age in only one tissue). The spleen and kidney exhibited the largest number of such proteins. Functional analysis revealed that kidney-specific downregulated proteins were enriched in pathways related to fatty acid oxidation and carboxylic acid metabolism, while spleen-specific downregulated proteins mapped to translation, RNA processing, and proteostasis maintenance pathways.

These results provide a comprehensive view of how aging affects tissues in both shared and divergent ways. While certain pathways, such as immune activation, appear to be broadly upregulated with age, others, such as mitochondrial metabolism and proteostasis, show tissue-dependent regulation, potentially reflecting organ-specific resilience or vulnerability to aging stressors.

## Highly correlated expression of non-blood DEPs shared across tissues

Towards identifying shared aging-associated dysregulation of proteins across tissues, we probed for trajectory correlation of

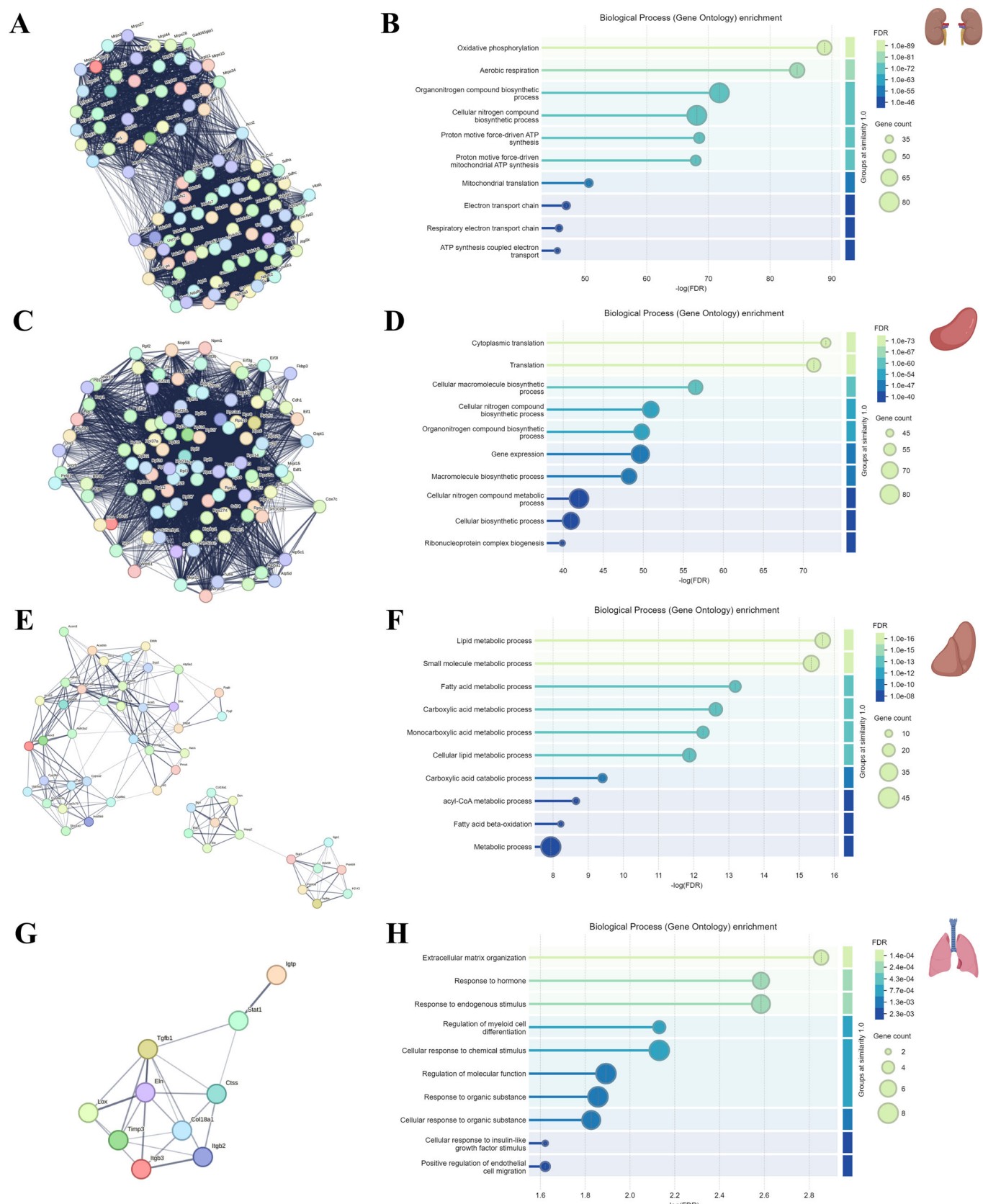

**Figure 5. Networks and gene ontology analysis of the top protein hubs associated with the non-blood-derived differentially expressed proteins of the kidney, spleen, liver and lung.**

Kidney (**A, B**), spleen (**C, D**), liver (**E, F**) and lung (**G, H**). Non-blood proteome-derived DEPs were input into Cytoscape to determine the top 100 (kidney and spleen), 50 (liver), or 10 (lung) protein hubs based on maximal clique centrality (MCC). Networks for the top protein hubs and corresponding GO enrichment analyses of the kidney (**A, B**), spleen (**C, D**), liver (**E, F**), and lung (**G, H**), were visualized in STRING. Nodes represent proteins and edges indicate functional associations, highlighting highly connected nodes ("hubs"), with potential roles in coordinating functional networks of age-associated changes in each organ. Bubble size and color gradient represent the number of associated DEPs and FDR significance. Source data are available online for this figure.

protein changes amongst shared DEPs in kidney and liver, kidney and spleen, as well as the liver and spleen, using Spearman's rank correlation coefficient (Spearman, 2010). Expression patterns of carboxylic acid catabolism-associated proteins (AADAT, ACADSB, DCXR, ECI1, ETFDH, and HIBADH) and fatty acid metabolism-associated proteins (ALDH3A2, ACADS, ACADSB, ACSM3, and GGT5) were highly correlated in the kidney and liver (Fig. EV4; Dataset EV10). The proteins exhibited mostly stable expression between 3 and 14 months, followed by a decrease between 14 and 20 months, then a reversal to earlier levels from 20 to 26 months during aging. This pattern suggests age-associated shifts in carboxylic acid catabolism and acyl-CoA metabolism, which may reflect metabolic remodeling in both organs. In the kidney and spleen, we found that proteins involved in cellular energy metabolism, including ATP synthase subunits (ATP5PB), Complex I (NDUFA2), Complex II (SDHA), and oxidative phosphorylation-related proteins (COX5B, COX6B1, COX6C, UQCRC2), were highly correlated across these tissues (Fig. EV4; Dataset EV11). In addition, PDHA1, which links glycolysis to the TCA cycle, and ETFB, which facilitates electron transfer to ubiquinone, showed strong co-expression patterns, further highlighting the coordinated regulation of mitochondrial function. While AK2 contributes to ATP homeostasis, and GUCY1B1 participates in cGMP signaling, their correlations may reflect broader metabolic network interactions rather than direct involvement in oxidative phosphorylation (Fig. EV4; Dataset EV11). Their expressions were mostly stable between 3 and 14 months, followed by a decline between 14 and 20 months, before a reversal to earlier levels at 26 months. Alterations in cellular respiration-associated proteins may reflect age-associated metabolic adaptations, which could impact mitochondrial efficiency in the kidney and spleen. Mitochondrial function is known to decline with aging, yet these shifts may also represent compensatory responses to changes in nutrient availability. Additionally, we observed a strong correlation in the expression of ECM proteins (BGN, DCN, HSPG2, and SDC4) across the liver and spleen, with relatively stable levels between 3 and 8 months, an increase between 14 and 20 months, and a subsequent decline from 20 to 26 months (Fig. EV4; Dataset EV12), suggesting a shared age-associated remodeling of the ECM. The most highly correlated proteins across the three organ pairs showed enrichment in fatty acid oxidation-related pathways (HADHB, ECI1), carboxylic acid catabolism (HADHB, SCP2, DCXR, ECI1, and HIBADH), and ECM organization (TGFBI, FBLN1, HSPG2, ELN, COL18A1, and EMILIN1) (Fig. EV5; Dataset EV13). SCP2 plays a role in cholesterol metabolism and may be localized to peroxisomes, its co-expression with metabolic proteins suggests a broader coordination of age-associated metabolic remodeling across tissues.

## Targeted-MS-based validation confirmed blood carrier proteins, complement factors and ECM proteins/modifiers as DEPs

Based on the results of our global proteomic profiling of mouse organs, which indicated that most age-associated protein changes occur in the kidney, spleen, liver, and lung, we performed SureQuant-based quantitative MS analysis on a subset of differentially expressed proteins from these four organs, derived from animals aged 3, 5, 8, 14, or 20 months. The 26-month-old mice were excluded from validation experiments since the age group constituted only five biological replicates and the animals could be compromised by survivorship bias (Xie et al, 2022). Candidate proteins were chosen based on differential expressions in the moderated F-test analysis and had to be shared across 3 or more organs. The selected candidate target proteins for validation primarily consisted of blood carrier proteins, complement factors and ECM proteins/modifiers. We validated 23–36 and 16–22 target peptides and proteins, respectively in the 4 organs (Fig. 6A,B; Appendix Figs. S1–4). Several validated target peptides/proteins exhibited significant differential expressions, dominant in certain organs. Dysregulation of cathepsin D (CTSD), alpha-galactosidase A (GLA), kallikrein-1 (KAL-B), and SPARC-like protein 1 (SPARCL1) was dominant in kidney derived from 20-month-old mice (Appendix Fig. S1). Finally, the spleen exhibited changes in interferon alpha/beta receptor 2 (IFN-R-2) (Appendix Fig. S2). In the liver, we detected protein changes in basement membrane-specific heparan sulfate proteoglycan core protein (HSPG2) and decorin (PG-S2) (Appendix Fig. S3) at the same age. In the lung of 20-month-old mice, notable dysregulation was observed for Alpha-1-acid glycoprotein 2 (AGP 2), epidermal growth factor receptor (EGFR), solute carrier family 2, facilitated glucose transporter member 3 (GLUT-3) and cytochrome b-245 heavy chain (GP91-1) (Appendix Fig. S4).

Most validated target peptides/proteins were common across two, three, or all four organs. For example, fetuin-B (FETUB), fibrinogen alpha chain (FGA), mannose-binding protein C (MBP-C), and pigment epithelium-derived factor (PEDF) were significantly expressed in both, kidney and spleen (Appendix Figs. S1 and 2), whereas beta-glucuronidase (GUSB), lysosomal alpha-mannosidase (LAMAN), and biglycan (PG-S1) demonstrated significant differential expression in the liver and spleen (Fig. 6B; Appendix Figs. S2 and 3). Clusterin (APO-J) and serotransferrin (TF) were differentially expressed in the lung/kidney and lung/spleen, respectively. Lactadherin (MFGM) was significantly upregulated in the lung, liver, and kidney (Appendix Figs. S1–4).

Augmented protein expressions of complement component C9 (C9), fibronectin (FN), and syndecan-4 (SYND4) were common among the liver, kidney, and spleen (Appendix Figs. S1–3). In

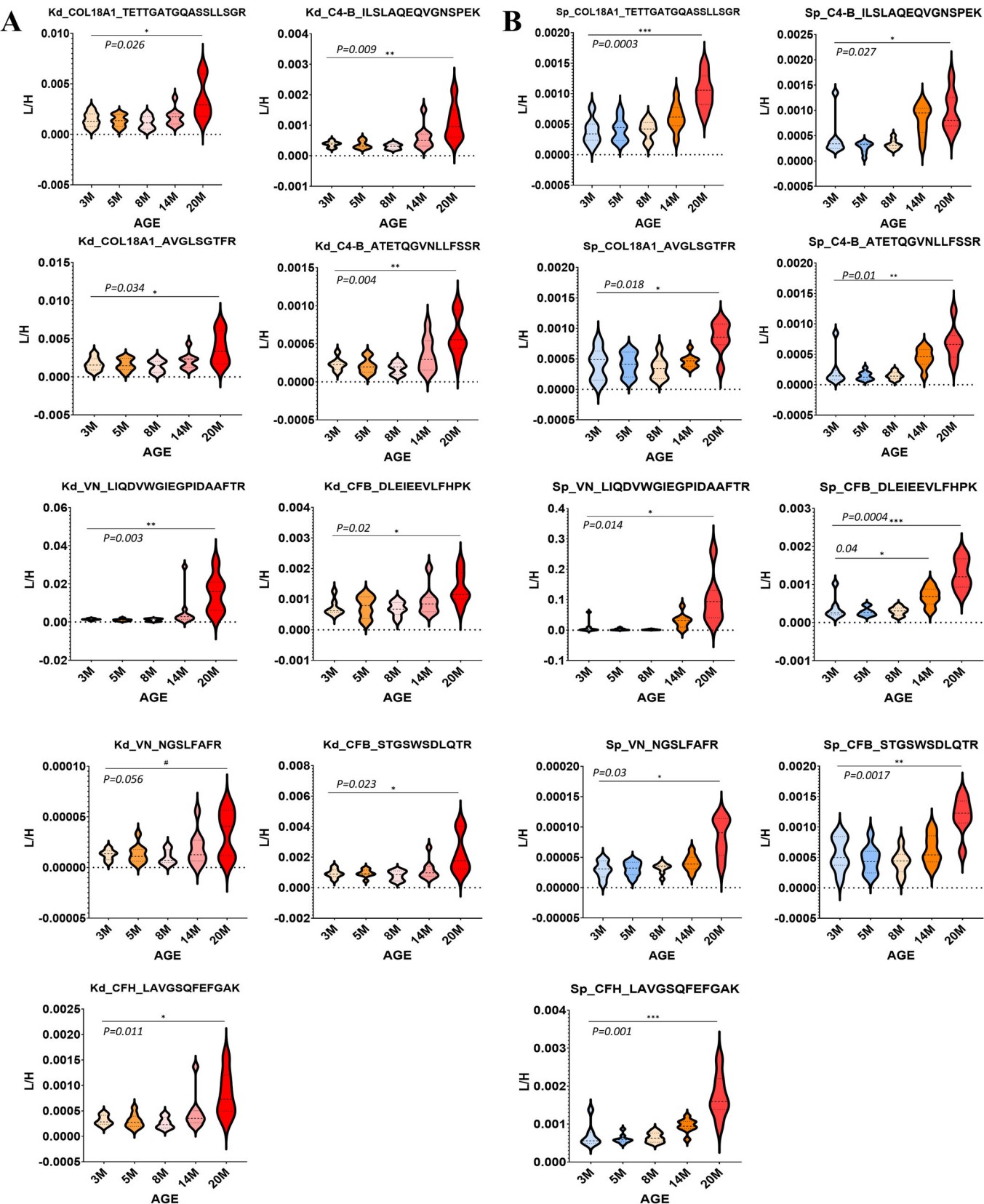

**Figure 6. SureQuant-based quantitative mass spectrometry analysis of shared ECM proteins and complement factors in the kidney and spleen.**

Comparison in the kidney (**A**) and spleen (**B**) was performed using the 3-month-old group as a reference and 7–9 biological replicates for each group. Mean light/heavy ratios (L/H) of unique peptides used for quantitation are depicted as violin plots. Bold dashed horizontal lines indicate medians, and lightly dashed lines indicate quartiles ($n = 3$). Statistical analysis was performed using the parametric unpaired one-sided $t$ test. The exact $P$ values are shown in each figure. $P < 0.05$ is statistically significant. Kd kidney, Sp spleen. 3, 5, 8, 14, and 20 M denotes age in months. Source data are available online for this figure.

addition, alpha-1-acid glycoprotein 1 (AGP-1) and von Willebrand factor A domain-containing protein 1 (VWA1) showed differential expressions in the lung, kidney, and spleen (Appendix Figs. S1–2 and S4). Notably, five validated target proteins displayed differential expression across all four organs, including complement factors (C4-B, CFB, and CFH) as well as ECM proteins (collagen alpha-1(XVIII) chain (COL18A1) and vitronectin (VN) (Fig. 6A,B).

## Discussion

In this study, we implemented an unbiased, quantitative global proteomics approach to investigate age-associated protein changes across 8 key organs in male C57BL/6J mice at six distinct time points (3, 5, 8, 14, 20, and 26 months). These ages span the young adult to late adult life stages, allowing us to capture age-associated protein changes in various mouse organs across much of the murine lifespan. Our proteomic analysis covered the brain, heart, kidney, liver, lung, skeletal muscle, spleen, and testis (Fig. 1A). Six of these organs overlap with those examined in recent studies by Keele et al (2023) and Oliviero et al (2022), creating a basis for comparative insights across investigations (Keele et al, 2023; Oliviero et al, 2022).

While Oliviero et al (2022) also assessed temporal protein changes across organs, their focus was primarily on the chromatin proteome, contrasting with our whole-tissue approach (Oliviero et al, 2022). Keele et al (2023) similarly explored protein dynamics but concentrated on two time points, assessing adult and late midlife stages (Keele et al, 2023). More recently, Takasugi et al (2024) employed TMT-based MS analysis of whole-tissue lysates in 8 organs, examining protein changes from young adult into old age (Takasugi et al, 2024). A key differentiator in our study is the experimental design, which leverages a relatively large sample size across 6 time points, providing a comprehensive overview of protein changes throughout aging.

To gain deeper insight into age-associated proteomic changes without the constraints of pairwise comparisons, we employed the moderated F-test, which captures overall protein dynamics across time. Traditional pairwise comparisons between two time points can overlook the continuous nature of biological aging, potentially leading to biased interpretations (Ritchie et al, 2015). By comparing pairwise analyses (20-month vs. 3-month reference) with the moderated F-test in the kidney, liver, lung, and spleen (Fig. 1D,E; Dataset EV1), we found that pairwise comparisons underestimated the number of differentially expressed proteins (DEPs). Consequently, the moderated F-test proved more suitable for detecting progressive proteomic alterations across age. To further resolve age-associated molecular changes, we applied trajectory analysis to track protein expression dynamics over time. In addition, we dissociated blood-derived from non-blood-derived proteins, refining organ-dominant signatures. This approach enabled us to

identify both organ-dominant and shared age-associated protein signatures, providing a clearer, system-wide perspective on aging's impact across tissues.

Most protein changes measured across the various mouse organs in our study peaked at 20 months, with considerable overlap of differentially expressed proteins in 4 organs: kidney, spleen, liver and lung. A majority of the 39 shared DEPs across the 4 organs were blood-derived and primarily constituted of blood carrier proteins, coagulants, immunoglobulins, complement factors and serpins. After exclusion of the blood-derived proteins, the remaining shared DEPs from the non-blood proteome comprised 8 proteins, primarily involved in immune regulation and antigen processing (LGALS3BP, STAT1 and TAPBP) (Teng et al, 2002; Trahey and Weissman, 1999; Nair et al, 2002) or ECM remodeling (COL18A1 and ELN) (Shimshoni et al, 2021). Our findings are in contrast to most previous reports, which indicated only modest alterations in protein levels of the mouse organs studied, at the late adult stage (Angelidis et al, 2019; Oliviero et al, 2022; Ori et al, 2015; Walther and Mann, 2011). The majority of these studies, either involved relatively small sample sizes with a few organs analyzed or included only two age groups for comparison, which may partially account for the discrepancies between their findings and our results reported herein (Angelidis et al, 2019; Ori et al, 2015; Walther and Mann, 2011).

Trajectory analysis of age-associated DEPs across the four organs with the most significant changes indicated that the onset and progression of aging manifests distinctly at the molecular level within each organ. In the kidney and liver, significant shifts in DEPs emerged between 14 and 20 months, with declining and minimally reduced protein expression at 26 months, respectively (Fig. 2A,C). Conversely, in the spleen and lung, these age-related molecular changes become evident at an earlier stage, between 8 and 14 months, peaking at 20 months. Following this peak, the changes in the spleen appear to diminish, while the lung shows sustained protein expression changes at 26 months (Fig. 2B,D). This early onset of significant changes in the spleen and the lung suggests that aging processes may initiate earlier in these organs compared to others, offering an organ-specific insight into the temporal dynamics of aging. In the kidney, mitochondrial proteins involved in electron transport and ATP synthesis (cluster 3) exhibited stable expression from young adulthood to midlife, followed by a sharp decline from 14 to 20 months and a partial recovery from 20 to 26 months (Fig. 3A). This decline coincided with reductions in TFAM (mitochondrial biogenesis) (Kelly and Scarpulla, 2004), SOD2 (antioxidant defense) (Cramer-Morales et al, 2015), and PARK7 (mitochondrial protection) (Irrcher et al, 2010), suggesting a transient phase of mitochondrial vulnerability (Dataset EV4). The subsequent recovery of VDAC1, TFAM, and SOD2 at 26 months indicates a compensatory mitochondrial remodeling response, potentially reflecting metabolic adaptation (Vander Heiden et al, 2000). Alternatively, the apparent recovery

detected at the population level may result from differential survivorship, as approximately 30% of the mouse population is lost to natural death between 20 and 26 months of age (Xie et al, 2022). In the aging spleen, both clusters, 1 and 2, are prominent. Cluster 1, enriched with ribosomal and mitochondrial proteins, showed stable expression throughout young adult stages, followed by a rapid decline between 8 and 20 months and an apparent partial recovery between 20 and 26 months. This drop indicates impaired protein synthesis and mitochondrial function, while the late recovery may suggest that the spleen attempts to restore these essential processes. Cluster 2, mainly containing ECM proteins, demonstrated a gradual increase between 8 and 20 months, followed by a partial decline from 20 to 26 months (Fig. 3B). The accelerated upregulation may signify ECM remodeling in response to tissue aging. In the liver, cluster 4 enriched with proteins involved in fatty acid and energy metabolism—displayed stable expression during young adult stages, gradually declining from 8 to 20 months and stabilizing thereafter (Fig. 3C). This trend aligns with a metabolic decline during aging. Notably, in the lungs, none of the three DEP clusters showed clear enrichment for specific protein groups (Fig. 3D), indicating a potentially distinct aging profile in this organ.

Network analysis in Cytoscape identified core protein hubs within each organ, allowing us to map connectivity among proteins subjected to age-dependent change. This focused approach across multiple time points and organs sets our study apart, revealing organ-dominant aging signatures rather than broad systemic signals, as seen in studies like Takusagi et al (Takasugi et al, 2024). In the kidney, a substantial number of mitochondrial electron transport chain components, including: 28 out of 45 Complex I subunits (e.g., NDUFS3, NDUFA1, NDUFA2, NDUFS4, NDUFB9, NDUFS2, NDUFV2), 3 out of 4 Complex II subunits (SDHA, SDHB, SDHC), 6 out of 11 Complex III subunits (CYC1, UQCRB, UQCR10, UQCRFS1, UQCRC1, UQCRC2), 7 out 13 Complex IV subunits (COX5A, COX6B1, COX4I1, COX7C, COX6C, COX7A2, COX7A1) and 11 out of 17 Complex V subunits (ATP5C1, ATP5O, ATP5D, ATP5PB, ATP5H, ATP5L, ATP5J2, ATP5K, ATP5E, ATP5A1, ATP5MD), were among the top-ranked hubs (Fig. 5A; Dataset EV7), reflecting the potential role of aging on mitochondrial bioenergetics (Althoff et al, 2011; Pereira et al, 2020). In addition, mitochondrial ribosomal proteins (e.g., MRPS2, MRPS16) were prominent hubs (Fig. 5A; Dataset EV7), emphasizing the central role of mitochondrial energy metabolism and protein synthesis in kidney aging (Miller et al, 2004). Their identification as network hubs reinforces the kidney's reliance on mitochondrial function and suggests that these proteins could serve as early biomarkers or therapeutic targets in preventing renal aging. EIF3I and RPS27, which are involved in the initiation of protein translation and assembly or function of ribosomes, emerged as key hubs in the spleen (Fig. 5B; Dataset EV7) (Vladimirov et al, 1996). Their presence suggests that the regulation of protein synthesis is central to spleen aging. In the liver, lipid metabolism hubs like ACLS1 and HMGCS2 indicate that energy-processing proteins are pivotal in liver aging (Fig. 5C; Dataset EV7) (Asif et al, 2022). Age-dependent change in ACLS1 has implications for triacylglycerol synthesis, beta-oxidation and phospholipid fatty acid composition (Li et al, 2009). In addition, protein changes in HMGCS2 are associated with aberrant ketogenesis and subsequent development of fatty liver disease (Asif et al, 2022).

By pinpointing these as central nodes, our study refines the understanding of metabolic aging in the liver, suggesting that targeted support to these hubs may maintain hepatic function in advanced age. The ECM-associated hubs in the lung, such as COL18A1 and ELN, emphasize the importance of structural proteins in pulmonary aging (Fig. 5D; Dataset EV7) (Devarajan et al, 2023). These proteins form a network that governs ECM stability and late-stage upregulation of COL18A1 and downregulation of ELN, both of which highlight a unique pattern of structural vulnerability to lung fibrosis (Karsdal et al, 2017) and reduced elasticity (Mecham, 2018) in the aging lung. The identification of specific ECM hubs offers potential targets for prolonging lung function by maintaining tissue elasticity.

Gene ontology analysis of the protein networks for the non-blood DEPs of the four organs (kidneys, spleen, liver, and lungs) with the most connections revealed enrichment of the following organ-dominant biological processes: oxidative phosphorylation (kidney), cytoplasmic translation (spleen), lipid metabolic process (liver) and extracellular matrix organization (lung) (Fig. 5). We evaluated the functional enrichment of GO terms within the non-blood DEPs that were shared across 2 or more organs using network analysis in STRING and determined that mitochondrial electron transport chain (ETC) and ATP production, as well as oxidative stress and antioxidant defense, were highly represented in the kidney and spleen (Fig. EV3; Dataset EV4). The mitochondrial and oxidative stress-associated proteins were downregulated in both organs during normal aging. In contrast, the kidney- and liver-dominant DEPs showed enrichment in fatty acid and energy metabolism, as well as proteins associated with extracellular matrix (ECM) remodeling (Fig. EV3; Dataset EV4). The former were downregulated, whereas the latter were upregulated in both. Similarly, the shared spleen- and liver-associated non-blood DEPs showed overrepresentation of proteins linked to immune regulation and antigen presentation, as well as extracellular matrix (ECM) remodeling and fibrosis (Fig. EV3; Dataset EV4). Moreover, the shared kidney-, spleen- and liver-associated DEPs were enriched in ECM organization and regulation of cytoskeleton organization (Fig. EV3; Dataset EV4). Both immune regulation/antigen presentation and cytoskeleton organization-associated proteins were upregulated in the liver and downregulated in the spleen (Dataset EV2). This suggests that normal aging in the kidney and spleen is characterized by reduced mitochondrial function and impaired ATP production, increased oxidative stress, and consequently reduced cellular energy homeostasis. Dysregulated lipid metabolism and energy homeostasis was a feature of the aging kidney and liver, whereas alterations in ECM remodeling and cytoskeletal dynamics were hallmarks of the aging kidney, spleen and liver. In addition, immune activation was associated with aging in the liver and spleen. Protein changes in the spleen were first detectable at 8 months, whereas they emerged at 14 months in the kidney, liver and lung (Figs. 2A–D and 3A–D). These findings provide evidence for aging-associated biological processes which are either prominent in specific organs compared to others (e.g., altered mitochondrial function in kidney and spleen) or pervasive across multiple organs (e.g., ECM remodeling and cytoskeletal dysregulation).

In conclusion, the study provides a novel, organ-specific, and temporally resolved understanding of protein dynamics during aging, offering unique insights into the biological processes and

potential intervention points that could support healthy aging. Our analysis revealed that age-associated protein changes proceed rapidly in the kidney and spleen, moderately in the liver and lung, minimally in the skeletal muscle and testis, but very subtly in the heart and brain. By isolating organ-dominant pathways and protein hubs, our analysis not only highlights distinct aging mechanisms but also underscores the varied magnitude and onset of the age-related protein changes in the different organs.

# Methods

### Reagents and tools table

| Reagent/resource | Reference or source | Identifier or catalog number |
|---|---|---|
| **Experimental models** | | |
| C57BL/6J mice | Jackson Laboratory | Stock no. 632 |
| **Recombinant DNA** | | |
| **Antibodies** | | |
| **Oligonucleotides and other sequence-based reagents** | | |
| **Chemicals, enzymes, and other reagents** | | |
| 4-(2-hydroxyethyl)-1-piperazineethanesulfonic acid (HEPES) (pH 7.4) | Thermo Scientific | 15630080 |
| Sodium chloride (NaCl) | Sigma Aldrich | S5150-1L |
| Ethylenediaminetetraacetic acid (EDTA) | Thermo Scientific | |
| Sodium Dodecyl Sulfate (SDS) | Roth | 2326.2-500 g |
| Dithiothreitol (DTT) | Thermo Scientific | R0862 |
| tris(2-carboxyethyl)phosphine (TCEP) | Thermo Scientific | 77720 |
| Iodoacetamide (IAA) | Thermo Scientific | 35603 |
| Pierce Protease and Phosphatase Inhibitor Mini Tablets,EDTA-free | Roche | A32961 |
| ammonium bicarbonate | Thermo Scientific | 393212500 |
| Potassium chloride (KCL) | Thermo Scientific | A11662.0B |
| AMICON ULTRA-0.5 CENTRIFUGAL FILTER UNI, 10 K MWCO, 0.5 ML (96 pack) | Millipore | UFC501096 |
| Eppendorf™ Protein LoBind | Thermo Scientific | 10708704 |
| SILVERQUEST KIT 1 KIT/STAINING FOR 25 MINI GEL | Sigma Aldrich | LC6070 |
| Synthetic PEPotec isotope-labeled C-terminal lysine (K) or arginine (R) crude peptides | Thermo Scientific | |
| PIERCE C18 TIPS, 100 UL BED 96 TIPS | | 87784 |
| Pierce™ LC-MS Grade water | Thermo Scientific | 85189 |
| 0.1% Formic acid (v/v) in water, LC-MS Grade | Thermo Scientific | 85171 |

| Reagent/resource | Reference or source | Identifier or catalog number |
|---|---|---|
| 0.1% Trifluoroacetic Acid (v/v) in water, LC-MS Grade | Thermo Scientific | 85173 |
| Pierce™ Acetonitrile, LC-MS Grade | Thermo Scientific | TS-51101 |
| Isopropanol, Optima™, LC-MS Grade | Thermo Scientific | A461-4 |
| Trifluoroacetic acid (TFA) | Thermo Scientific | |
| PIERCE TRYPSIN PROTEASE MS GRADE, 5 ×20 UG | Thermo Scientific | 90058 |
| Urea | Thermo Scientific | |
| Acclaim PepMap C18, 100 Å, 5 mm×300 μm | Thermo Scientific | 160454 |
| Acclaim PepMap C18, 100 Å, 75 μm X 50 cm | Thermo Scientific | 164942 |
| Stainl. Steel Emitters, 40 mm, OD 1/32 | Thermo Scientific | ES542 |
| Pierce™ FlexMix™ Calibration Solution | Thermo Scientific | A39239 |
| LC-MS autosampler vials (CGC Certified Clear Glass 12 x 32 mm Screw Neck Total Recovery Vial, with Cap and PTFE/silicone Septum, 1 mL Volume, 100/pkg | Waters | 186000384C |
| **Software** | | |
| Proteome Discoverer™ software (v3.1) | Thermo Scientific | OPTON-30904 |
| QFeatures (v1.12.0) | Open source | |
| Skyline software v21.1.0.278 | Open source | |
| R v4.3.1 | Open source | |
| Cytoscape | Open source | https://cytoscape.org/ |
| STRING | Open source | https://string-db.org/ |
| **Other** | | |
| RSLCnano system (Ultimate 3000) | Thermo Scientific | Serien-Nr.: 8163996 |
| Orbitrap Exploris 480 mass spectrometer | Thermo Scientific | Serien-Nr.: MA10019N |

## Animal housing and husbandry conditions

This study utilized male C57BL/6J mice (Jackson Laboratory, stock no. 632), divided into six distinct age cohorts: 3, 5, 8, 14, 20, and 26 months. All animals were acquired as a single batch and maintained under specific pathogen-free (SPF) conditions in individually ventilated cages (IVCs), adhering to FELASA guidelines. Each cage housed a maximum of five mice from the same age group. Environmental conditions were kept constant, with a temperature of 22 °C, 55% humidity, and a 12-h light/dark cycle. Mice had unrestricted access to food and water, in accordance with

institutional and governmental animal welfare regulations. Daily monitoring was conducted to ensure general animal well-being, and individual health assessments were performed before commencing any experimental procedures. These assessments followed criteria approved by the local animal ethics committee. Animals meeting predefined humane endpoints, such as visible bleeding or ulcerated tumors, were humanely euthanized. Sample size estimations for each age group were calculated using G*Power software (version 3.1.9.2), in alignment with ethical approval requirements.

### Sample preparation for mass spectrometry analysis

Prior to tissue collection, mice were acclimated in the dissection room for a minimum of 30 min. Mouse organs, including: the brain, spleen, lung, liver, kidney, testis, skeletal muscle and heart (Fig. 1) were harvested from male C57BL/6J mice over the course of two days and at the different time points, as previously described (Xie et al, 2022). The number of animals processed each day was evenly distributed across age groups. Tissues were rapidly frozen in liquid nitrogen and stored at $-80\,°C$. For protein extraction, frozen organs were cryopulverized under liquid nitrogen to yield fine, homogenous tissue powder, which was then used for peptide generation and high-resolution accurate mass (HRAM) spectrometry. Each sample was lysed in 200 μl of buffer composed of 50 mM HEPES (pH 7.4), 150 mM NaCl, 1 mM EDTA, 1.5% SDS, and 1 mM DTT, supplemented with a protease and phosphatase inhibitor cocktail (Thermo Scientific). Lysis was enhanced via repeated sonication cycles—six rounds of 1-minute water bath sonication at 35 kHz, interspersed with 2-minute cooling intervals on ice. Total protein content was estimated using silver staining prior to digestion. Approximately 25 μg of protein per sample was reduced and alkylated, then processed using a modified Filter-Aided Sample Preparation (FASP) protocol (Scifo et al, 2015) to generate tryptic peptides for subsequent label-free and SureQuant-based targeted proteomic analyses. FASP enables effective proteolysis of low-quantity samples, producing high-quality tryptic peptides for shotgun proteomics (Wisniewski, 2016; Wisniewski et al, 2009). Tryptic digestion was carried out overnight at 37 °C directly on the filter units using trypsin at a 1:20 enzyme-to-protein ratio in 50 mM ammonium bicarbonate. Residual detergents were precipitated by adding an equal volume of 2 M KCl. The resulting peptides were then purified and desalted using C18 StageTips and reconstituted in 20 μl of 1% formic acid for LC-MS/MS analysis. Mass spectrometry was performed on non-blinded but randomly ordered samples, with 5–9 biological replicates per condition.

### Liquid chromatography and tandem mass spectrometry (LC-MS/MS) analysis

Peptide mixtures were analyzed using a Dionex Ultimate 3000 RSLC nanoLC system coupled to an Orbitrap Exploris 480 mass spectrometer. Peptides were injected using a solvent consisting of 95% eluent A (0.1% formic acid in water) and 5% eluent B (0.1% formic acid in 80% acetonitrile) at a flow rate of 300 nL/min. Samples were first concentrated on a trap column (Acclaim PepMap C18, 100 Å, 5 mm × 300 μm i.d.; Thermo Scientific), followed by separation on an analytical column (Acclaim PepMap C18, 100 Å, 75 μm × 25 cm; Thermo Scientific) using reversed-phase chromatography. Peptides were eluted with a 75-min linear gradient from 5 to 31% eluent B, followed by a 20-min ramp up to 50% eluent B. Mass spectrometry was performed in positive ion mode with data-dependent acquisition. Full MS (MS1) spectra were collected at 120,000 resolution over an *m/z* range of 375–1550, using an AGC target of $3 \times 10^6$ ions, maximum injection time (maxIT) of 25 ms, and a charge state range of 2 to 7. Dynamic exclusion was set to 60 s with single-event exclusion and a mass tolerance of 10 ppm. For MS/MS (MS2), precursor ions were selected using a top-speed approach with a 2-s cycle time. A decision tree strategy was employed to prioritize precursor ions: those with signal intensities above $3 \times 10^5$ were given highest priority (scan priority one), and those between $1 \times 10^4$ and $3 \times 10^5$ were assigned as scan priority two. MS2 settings included a 2 *m/z* isolation window, normalized collision energy (NCE) of 30% using higher-energy collisional dissociation (HCD), and resolutions of 7500 and 15,000 for priority one and two scans, respectively. AGC targets were set to $1 \times 10^5$ with maxITs of 20 ms and 50 ms. Full MS data were acquired in profile mode, while fragment ion spectra were recorded in centroid mode.

### Database searching

Mass spectrometry raw files were analyzed using Proteome Discoverer™ software (version 3.0.1.27, Thermo Scientific), employing the SEQUEST® HT algorithm to search against the Mus musculus Swiss-Prot® protein database (release date: 2023-11-08). Tryptic digestion was assumed with up to two missed cleavages permitted, and peptide lengths were restricted to between 7 and 30 amino acids. Precursor ion mass tolerance was set at 10 parts per million (ppm), and fragment ion tolerance for MS2 was set at 0.02 Da. Carbamidomethylation of cysteine residues was specified as a fixed modification, while oxidation of methionine and N-terminal methionine loss were considered variable modifications. False discovery rates (FDR) for both peptide and protein identifications were controlled at 1%, as defined by the Peptide and Protein Validator nodes within the Consensus workflow. Unless otherwise specified, default settings were applied for all nodes. The Spectrum Selector node was configured to include charge states from +2 to +6. INFERYS rescoring was enabled in automatic mode, and final peptide-spectrum matches were filtered to a maximum of 1% FDR using the Percolator algorithm during the Processing workflow. A second-pass search was enabled to detect semi-tryptic peptides. For quantification, both razor and unique peptides were used. Prior to downstream analysis, entries flagged as reverse hits, identified by site only, or classified as potential contaminants were excluded.

### Data processing

After initial processing in Proteome Discoverer, peptide-level quantification data were exported and further curated using the QFeatures package (version 1.12.0) in R (version 4.3.1). Several filtering criteria were applied to ensure data quality: peptides lacking a corresponding master protein accession (Master.Protein.Accessions) were excluded; entries with quantification flags marked as "NoQuanValues" or "NoneMonoisotopic" (Quan.Info) were removed; only peptides linked to a single protein group (X..Protein.Groups == 1) and classified as unambiguous (PSM.Ambiguity == "Unambiguous") were retained. Additional filtering was

performed based on posterior error probabilities and q-values, both required to be ≤0.05 (as computed by Qvality). Peptide intensities were normalized using median centering (center.median) and log-transformed to stabilize variance and approximate normality. Protein-level quantification was then derived by aggregating peptides using the robustSummary method from the MsCoreUtils package, grouping by the master protein accession identifier. During aggregation, missing values were excluded (na.rm = TRUE). Remaining missing protein intensities were imputed using the missForest algorithm (version 1.5), a non-parametric random forest-based approach that iteratively predicts missing values based on observed features, offering robust imputation across complex datasets.

To detect differentially expressed proteins (DEPs), a moderated F-test was applied, enabling the evaluation of statistical differences across all experimental groups without requiring a predefined reference condition. This method assesses whether any condition differs from the others by considering all model coefficients simultaneously, making it especially appropriate for exploratory analyses with balanced group designs. The limma package (version 3.58.1) was used to fit a linear model to the protein expression data (lmFit), followed by empirical Bayes moderation to shrink variance estimates and enhance statistical reliability. An overall F-statistic and associated $P$ values were computed via the topTable function. Proteins with a Benjamini–Hochberg adjusted $P$ value < 0.05 were classified as significant. This approach provides a robust framework for identifying DEPs while accounting for variability across age groups and organs.

## Dimensionality reduction of samples in various mouse organs

To assess intra-organ variability across the six age points (3, 5, 8, 14, 20, and 26 months), we applied Uniform Manifold Approximation and Projection (UMAP) (Becht et al, 2019) (UMAP) to our proteomic datasets (Figs. 1B and 2E–H). Input data for UMAP consisted of log-transformed, normalized protein abundances from all age groups across the eight analyzed organs: brain, heart, kidney, liver, lung, skeletal muscle, spleen, and testis. For organ-specific dimensionality reduction, the full set of identified proteins per organ was used. For cross-organ visualization, we restricted the analysis to the 1,223 proteins commonly detected across all organs, as determined using the merge function in R. We evaluated three distance metrics "euclidean", "canberra", and "cosine" to determine which provided the clearest cluster separation. Since all three metrics produced comparable clustering results, we opted to use the default "Euclidean" metric for the final UMAP visualizations.

## UpSet plot analysis

To visualize overlap and specificity of age-associated changes, we analyzed differentially expressed proteins (DEPs) identified in the eight mouse organs—brain, heart, lung, liver, kidney, spleen, skeletal muscle, and testis—by comparing protein expression at 20 months to the 3-month baseline. Only DEPs identified with medium to high confidence were included. Visualization was performed using the UpSetR package (version 1.4.0) in R (version 4.3.1). The resulting UpSet plot illustrates both organ-specific (single black dots) and shared (connected dots) protein expression

changes across organs (Fig. 4B), effectively highlighting the unique and overlapping DEPs among tissues.

## Trajectory analysis

To investigate age-related protein expression dynamics, we performed trajectory analysis using log-transformed, normalized abundance values from the proteomics datasets. For each organ, protein expression levels across time points were first standardized using z-score normalization via the scale function in R, and the resulting average z-scores were used as input for time-series clustering. Temporal expression patterns were grouped into 2 to 4 clusters per organ using the fuzzy c-means algorithm implemented in the Mfuzz package (version 2.62.0) in R (version 4.3.1). Clustering parameters were guided by the partition coefficient (PC) and partition entropy (PE), with the optimal fuzzifier (m) estimated using the mestimate function (Futschik and Carlisle, 2005; Kumar and E Futschik, 2007). To compare aging trajectories across the eight mouse organs—brain, heart, kidney, liver, lung, skeletal muscle, spleen, and testis—we overlaid the most enriched clusters from each organ into a combined plot. This approach enabled us to distinguish between organ-specific and shared temporal protein expression patterns associated with aging. To interpret the functional relevance of these clusters, we performed Gene Ontology (GO) enrichment analysis using STRING (https://string-db.org/), focusing on biological processes and molecular functions enriched in each cluster (Figs. EV1 and 2).

## Construction of protein network hubs

To identify central regulatory proteins in aging-associated networks, we analyzed differentially expressed proteins (DEPs) from non-blood organs using Cytoscape (version 3.10.0; https://cytoscape.org/). For each organ, we selected the top-ranking hub proteins based on maximal clique centrality (MCC) using the CytoHubba plug-in. Specifically, the top 100 hubs were extracted for kidney and spleen, 50 for liver, and 10 for lung, based on network connectivity. Protein–protein interaction networks for these hub sets were then visualized using STRING, highlighting direct and indirect associations among the most connected proteins. Hub status was determined by the number of interactions each protein exhibited within the STRING-derived network.

## Linear mixed-effects modeling for tissue-specific aging effects

To detect proteins exhibiting tissue-specific expression changes with age, we employed a linear mixed-effects model (LMM) using the lme4 package in R. The model was specified as:

Expression ~ Age * Tissue + (1 | Mouse), where Age was treated as a continuous predictor, and a random intercept for Mouse accounted for intra-individual correlations arising from matched tissue samples. To test for tissue-specific age trajectories, we performed a likelihood ratio test (LRT) by comparing the full model (with the Age × Tissue interaction) to a reduced model without the interaction term. Proteins with a Benjamini–Hochberg adjusted $P$ value < 0.05 were considered to exhibit significant age-by-tissue interaction effects. For proteins identified as significant by the LMM, we further assessed the directionality of age-related

changes in each tissue by conducting linear trend tests. These were performed using ordinary least squares regression (lm(y ~ Age)), treating Age as a continuous variable. This analysis provided both the slope and statistical significance of expression changes over time for each protein-tissue pair. The results were summarized in a directionality matrix, categorizing proteins as being consistently upregulated, consistently downregulated, or showing divergent trends across tissues.

### Selection and characterization of isotope-labeled peptides for targeted mass spectrometry

Peptides for SureQuant-based targeted mass spectrometry (MS) were selected according to three main criteria. First, we prioritized unique peptides derived from candidate proteins identified as differentially expressed across age groups in our label-free proteomics dataset, based on moderated F-test results. Second, peptides with favorable ionization properties and previously observed in MS-based studies (as curated in PeptideAtlas) were preferred. Third, candidate peptides were required to be 8–25 amino acids in length and free of internal cysteine residues to ensure stability and detectability. Using these criteria, we acquired synthetic PEPotec peptides (Thermo Scientific) labeled with stable isotopes at the C-terminal lysine (K) or arginine (R). These crude, heavy-labeled peptides were pooled at equimolar concentrations (1 pmol/μL each in 0.1% formic acid) to create a spike-in standard for downstream MS analysis. Peptide intensity optimization was performed prior to the SureQuant assay. In the initial survey run, precursor ions were monitored using their monoisotopic mass within a 2–3 $m/z$ window, capturing reference fragment ions at a minimum intensity of $1 \times 10^5$. A second run was conducted using the precursor $m/z$ values identified in the first step to confirm and fine-tune fragment ion selection. For quantitative analysis, the same endogenous tryptic peptides from the label-free experiment were used, this time spiked with the stable isotope-labeled (SIL) counterparts. Target peptides quantified in the SureQuant assay are listed in Dataset EV14.

### Liquid chromatography and survey MS analyses

Stable isotope-labeled (SIL) and endogenous peptides were mixed and analyzed using a Dionex Ultimate 3000 RSLC nanosystem coupled to an Orbitrap Exploris 480 mass spectrometer. Samples were injected in a solvent containing 95% eluent A (0.1% formic acid in water) and 5% eluent B (0.1% formic acid in 80% acetonitrile), and loaded onto a trap column (Acclaim PepMap C18, 100 Å, 5 mm × 300 μm i.d., Thermo Scientific). Peptides were separated by reversed-phase chromatography on an analytical C18 column (75 μm × 25 cm, Thermo Scientific) using a 35-min linear gradient from 5% to 25% eluent B, followed by a 5-min ramp to 50% eluent B. Mass spectrometry was performed in positive ion mode using data-dependent acquisition (DDA). Instrument settings included a spray voltage of 1.9 kV, no sheath or auxiliary gas flow, and a capillary temperature of 300 °C. MS1 scans were acquired at 120,000 resolution, over a mass range of 300–1500 $m/z$, with an automatic gain control (AGC) target of $3 \times 10^6$ ions, a maximum injection time (maxIT) of 50 ms, and a default precursor charge state of 2. For each MS1 scan, the top 40 most intense ions

from the inclusion list exceeding an intensity threshold of $1 \times 10^5$ were selected for fragmentation. These were isolated within a 1.0 $m/z$ window and fragmented using higher-energy collisional dissociation (HCD) with a normalized collision energy (NCE) of 28%. MS2 spectra were recorded over a scan range of 100–1700 $m/z$, with an AGC target of $1 \times 10^7$ ions, maxIT of 10 ms, and a resolution of 7500.

### SureQuant-based quantitation of selected target peptides

SureQuant analysis was carried out following a described protocol (Stopfer et al, 2021), with specific adjustments tailored to our experimental design. Data acquisition was conducted using a modified SureQuant acquisition template, incorporating four branches to accommodate the +2 and +3 charge states of stable isotope-labeled (SIL) lysine- and arginine-containing peptides. Peak area data for endogenous (light) and corresponding heavy internal standard (IS) peptides were extracted using Skyline software (version 21.1.0.278) (MacLean et al, 2010). Only peptides for which the IS signals showed non-zero area under the curve (AUC) for at least five out of six monitored product ions were retained for analysis. For quantification, the three most intense fragment ions from both light and heavy peptides were selected. Their peak areas were summed, and the light-to-heavy ratio was used to calculate peptide abundance across samples. This approach—focusing on three high-intensity transitions—provided a compromise between quantitative precision and the ability to detect low-abundance targets.

### Quantification and statistical analysis

Statistical analysis details, including the tests and *N* number used, are provided in the figure legends.

## Data availability

Raw MS data, peak lists and search results are publicly available under the following accession numbers-ProteomeXchange (PXD058684) and jPOST (Okuda et al, 2017) (JPST003472) via the following link: https://repository.jpostdb.org/entry/JPST003472.3. Source data for Figs. 2, 3, EV and appendix Figs. have been deposited and is publicly accessible through the Data Dryad online repository under the following link: http://datadryad.org/share/WGg-0_wtEQp0kSMfDDtl5xdKmqR8Yu_9yRUGWZ2GCZ8. Analysis code is available at https://github.com/ehningerd/Scifo-Morsy-Liu_et_al-mouse_aging_proteome.

The source data of this paper are collected in the following database record: biostudies:S-SCDT-10_1038-S44318-025-00509-x.

## Peer review information

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

## Acknowledgements

Sarah Morsy is supported by the ETERNITY project consortium, which is funded by the European Union through Horizon Europe Marie Skłodowska-Curie Actions Doctoral Networks (MSCA-DN) under the grant number 101072759 (grant to Dan Ehninger). Ting Liu acknowledges financial support from the China Scholarship Council (CSC) (grant no. 202108080078).

## Author contributions

**Enzo Scifo**: Data curation; Formal analysis; Validation; Investigation; Methodology; Writing—original draft; Writing—review and editing. **Sarah Morsy**: Formal analysis; Investigation; Visualization; Methodology; Writing—original draft; Writing—review and editing. **Ting Liu**: Methodology. **Kan Xie**: Formal analysis; Methodology; Writing—review and editing. **Kristina Schaaf**: Methodology. **Daniele Bano**: Funding acquisition; Writing—review and editing. **Dan Ehninger**: Formal analysis; Supervision; Funding acquisition; Writing—original draft; Project administration; Writing—review and editing.

Source data underlying figure panels in this paper may have individual authorship assigned. Where available, figure panel/source data authorship is listed in the following database record: biostudies:S-SCDT-10_1038-S44318-025-00509-x.

## Funding

## Disclosure and competing interests statement

The authors declare no competing interests.

# Expanded View Figures

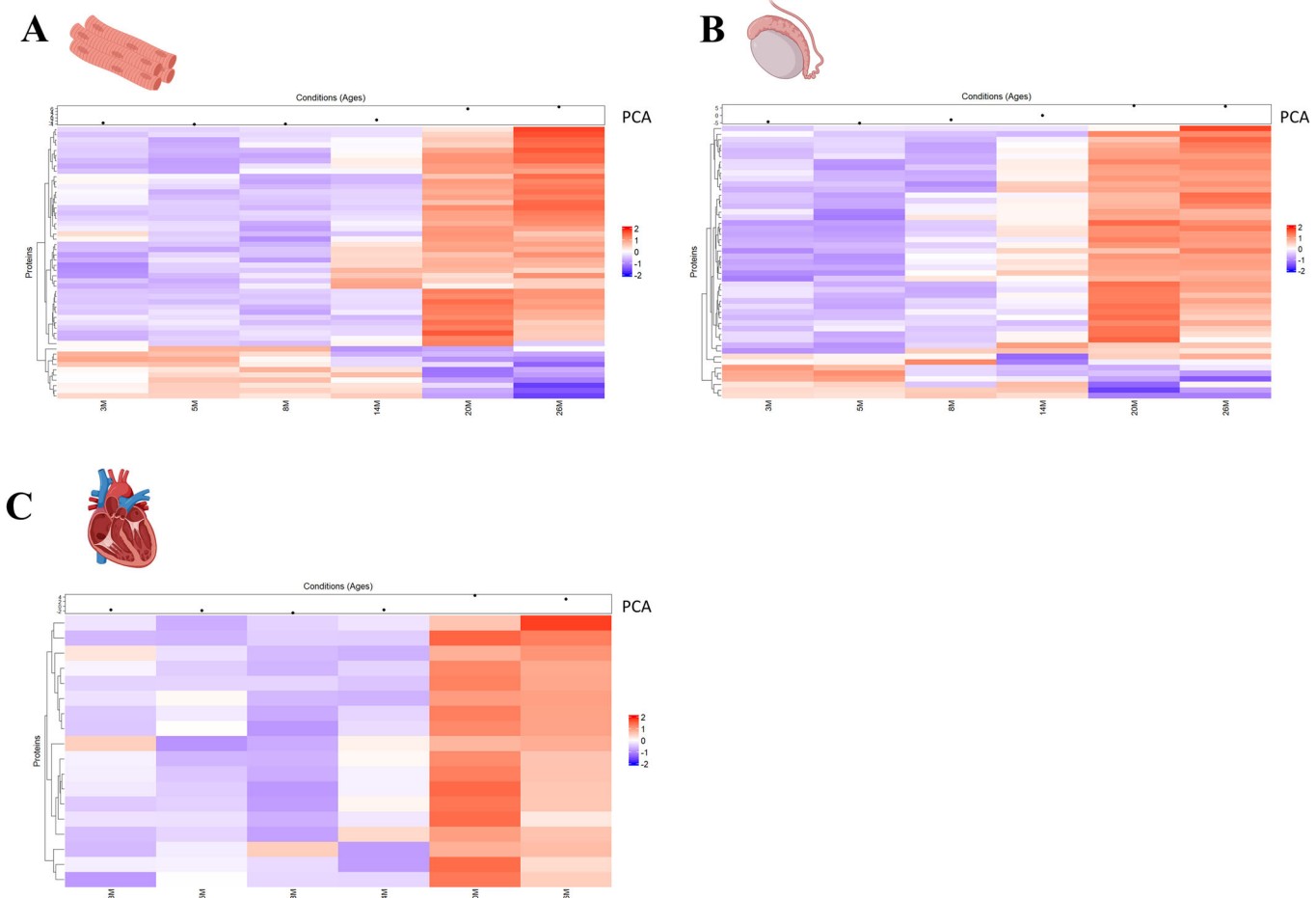

**Figure EV1.  Aging-associated protein expression changes in skeletal muscle, testis, and heart.**

(A–C) Heatmaps illustrate differentially expressed proteins (DEPs) across six age groups (3, 5, 8, 14, 20, and 26 months) for skeletal muscle (**A**), testis (**B**), and heart (**C**). Proteins (rows) are clustered by expression trends, while columns represent age groups. The color gradient reflects log-transformed expression values, with red indicating higher expression (upregulation) and blue indicating lower expression (downregulation) relative to each protein's average level. Line graphs adjacent to the heatmaps depict the mean z-scored expression trajectories over time, with the *x* axis representing age and the *y* axis showing expression changes. Dot plots at the top highlight the principal component 1 (PC1) trajectories, capturing age-related shifts in protein expression. PC1 accounts for 80%, 79%, and 83% of the variance in skeletal muscle, testis, and heart, respectively. Source data are available on the Data Dryad public repository for this figure.

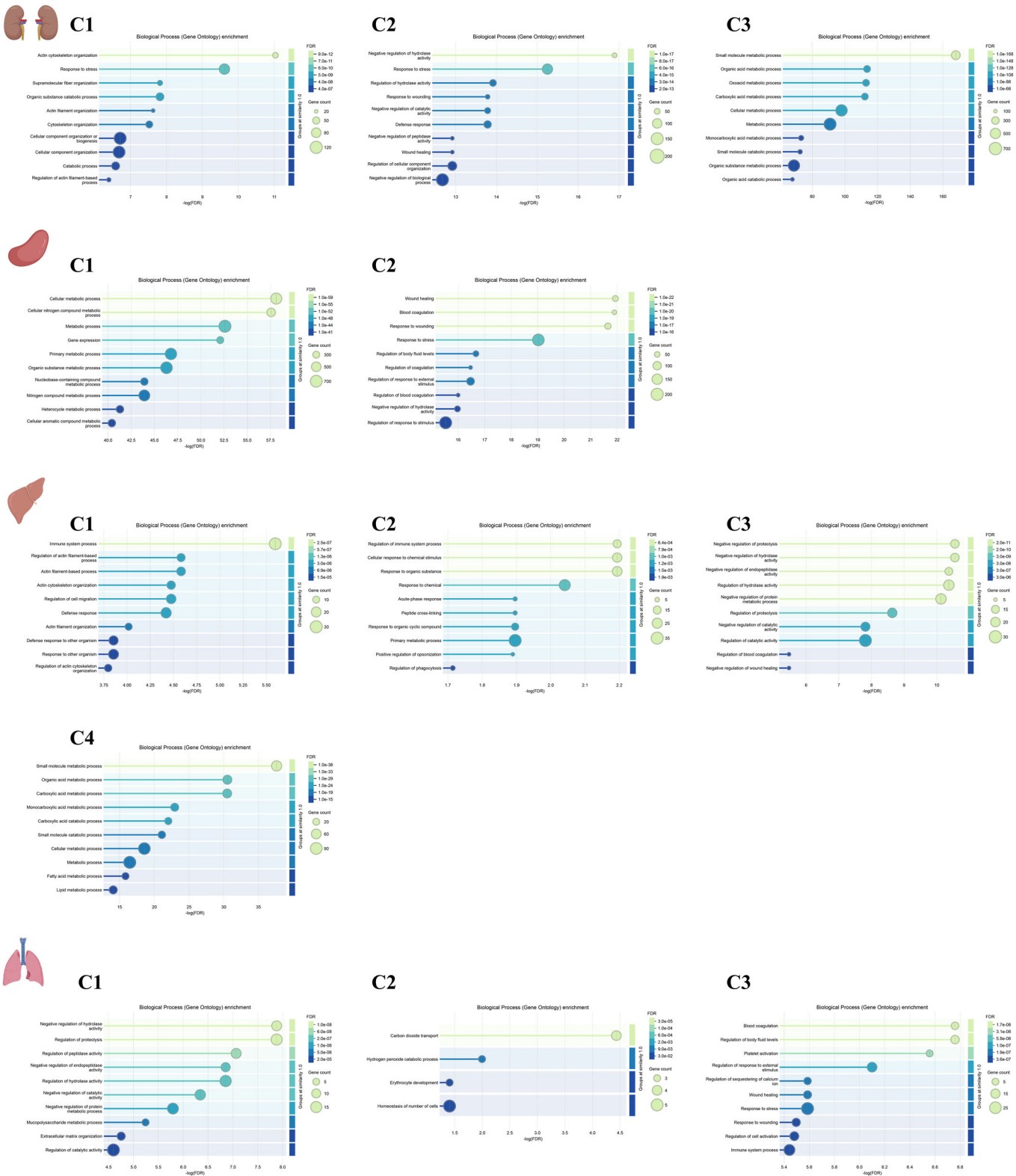

◄ **Figure EV2.  Gene Ontology annotation of individual enriched MFuzz clusters for age-DEPs including the blood-derived proteome in the kidney, spleen, liver, and lung.**

Enriched GO biological processes for clusters 1- 3 (C1-C3) for the kidney, clusters 1 and 2 (C1-C2) for the spleen, clusters 1–4 (C1-C4) for the liver and clusters 1–3 (C1-C3) for the lung are shown. Age-DEPs were clustered into temporal expression patterns using MFuzz clustering. Clusters represent groups of proteins with similar expression trajectories over six age groups (3, 5, 8, 14, 20, and 26 months). Gene Ontology (GO) enrichment analysis was performed for biological processes using STRING. The bar plots show enriched GO terms for each cluster, with the x axis representing -log10(P value) of enrichment and the y axis listing the GO terms. Bubble size indicates the number of proteins associated with each term, while the bubble color gradient corresponds to the false discovery rate (FDR), ranging from low (light green) to high (dark blue). Source data are available on the Data Dryad public repository for this figure.

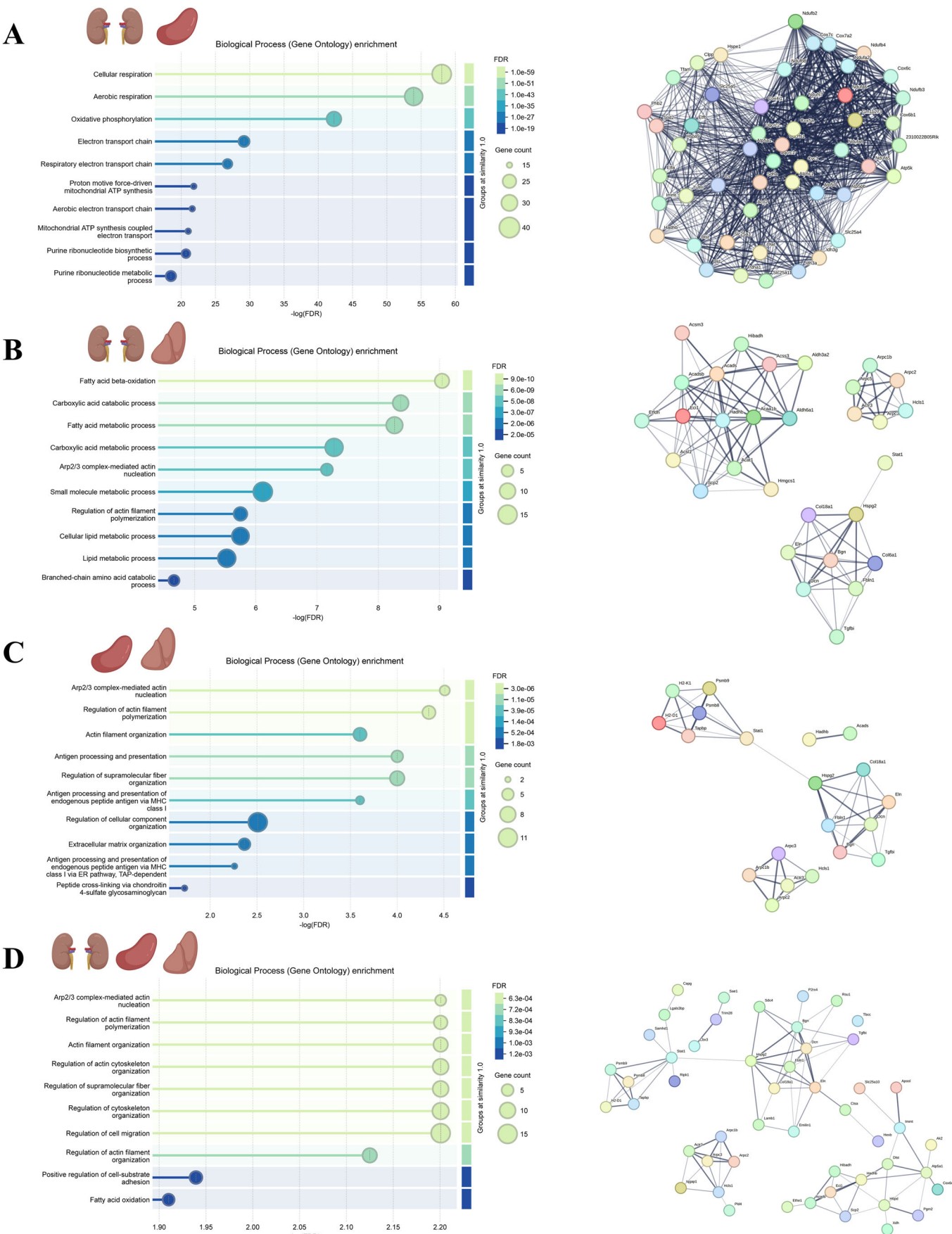

◀ **Figure EV3. Gene Ontology annotation and STRING network analysis of shared non-blood-derived age-DEPs in 2 or more organs.**

Enriched GO biological processes and STRING interaction network clusters for the top hub proteins identified in the shared age-DEPs between the kidney and spleen (**A**), kidney and liver (**B**), spleen and liver (**C**) as well as kidney, spleen and liver (**D**) are shown. Left Panels: GO plots showing enriched biological processes for the shared age-DEPs. The significance of enrichment is indicated on the x axis with the -log10(FDR), while protein count and false discovery rate (FDR), are depicted by the size and color of circles, respectively. Right Panels: STRING interaction networks for the top hub proteins identified using Cytoscape's CytoHubba. Each node represents a protein, and edges indicate functional associations. Source data are available on the Data Dryad public repository for this figure.

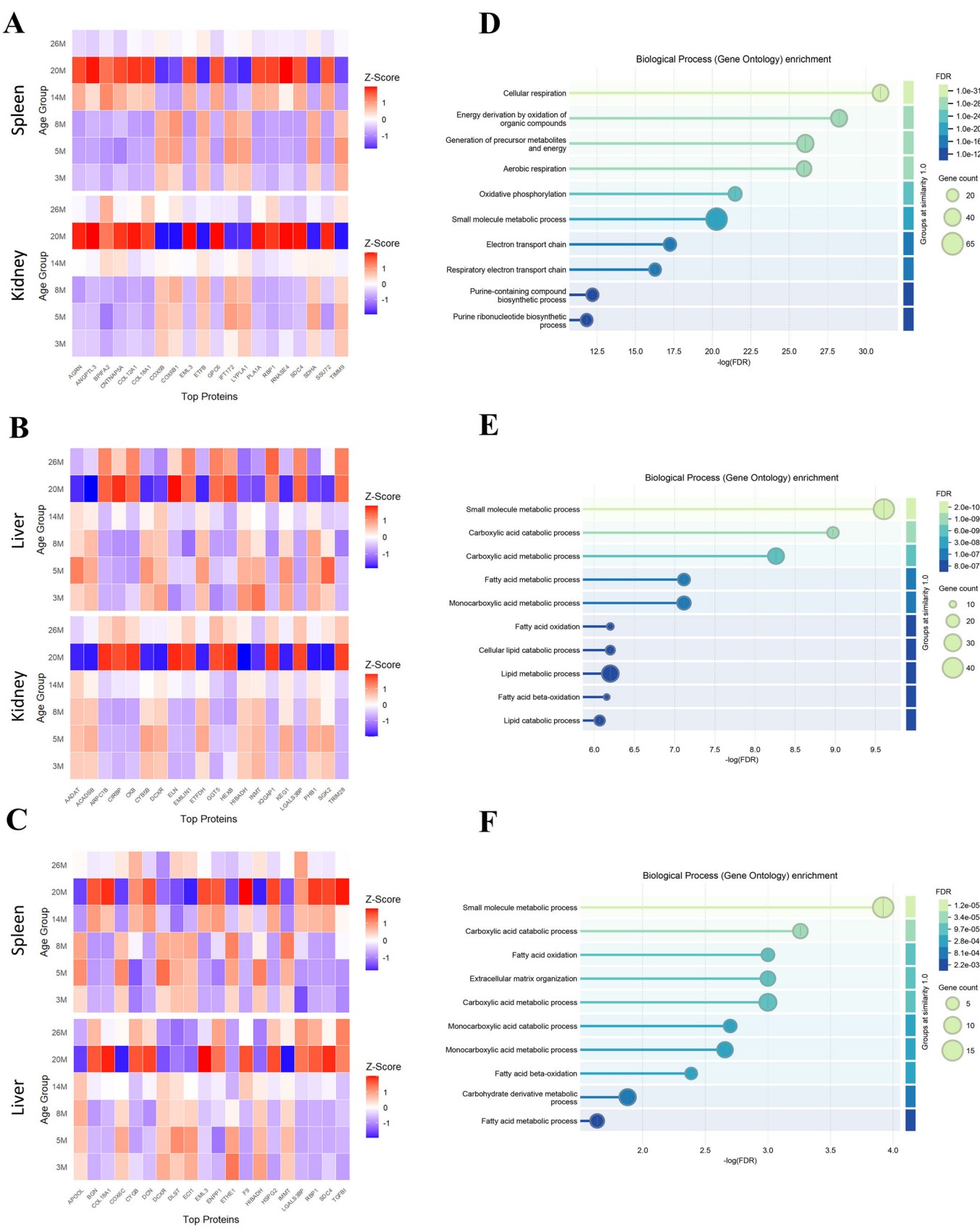

◀  **Figure EV4.  Pairwise correlation analysis of shared age-DEPs in kidney and spleen, kidney and liver, liver and spleen.**

Heatmaps displaying the expression trajectories of highly correlated proteins (Spearman's correlation coefficient >0.5) across paired organ comparisons: (A) spleen and kidney, (B) kidney and liver, and (C) spleen and liver. The color gradient indicates the normalized and log-transformed expression values, ranging from low (blue) to high (red) relative to the mean (standard error of the mean (SEM) is 0.1, 0.18, and 0.11 for the kidney, spleen, liver). Each heatmap represents the top 20 positively correlated proteins for the corresponding organ pair. Age groups (in months, e.g., 3 M, 5 M) are displayed on the y axis, and proteins are listed on the x axis. (D–F) Gene Ontology (GO) biological process enrichment analysis in STRING for proteins positively correlated (Spearman's correlation coefficient >0.5) in each organ pair: (D) spleen-kidney, (E) kidney–liver, and (F) spleen–liver. The x axis represents the -log10(FDR) of the enriched terms, while the bubble size corresponds to the number of genes associated with each term. The color gradient reflects the significance of enrichment, with lighter and darker shades indicating lower and higher FDR values, respectively. Source data are available on the Data Dryad public repository for this figure.

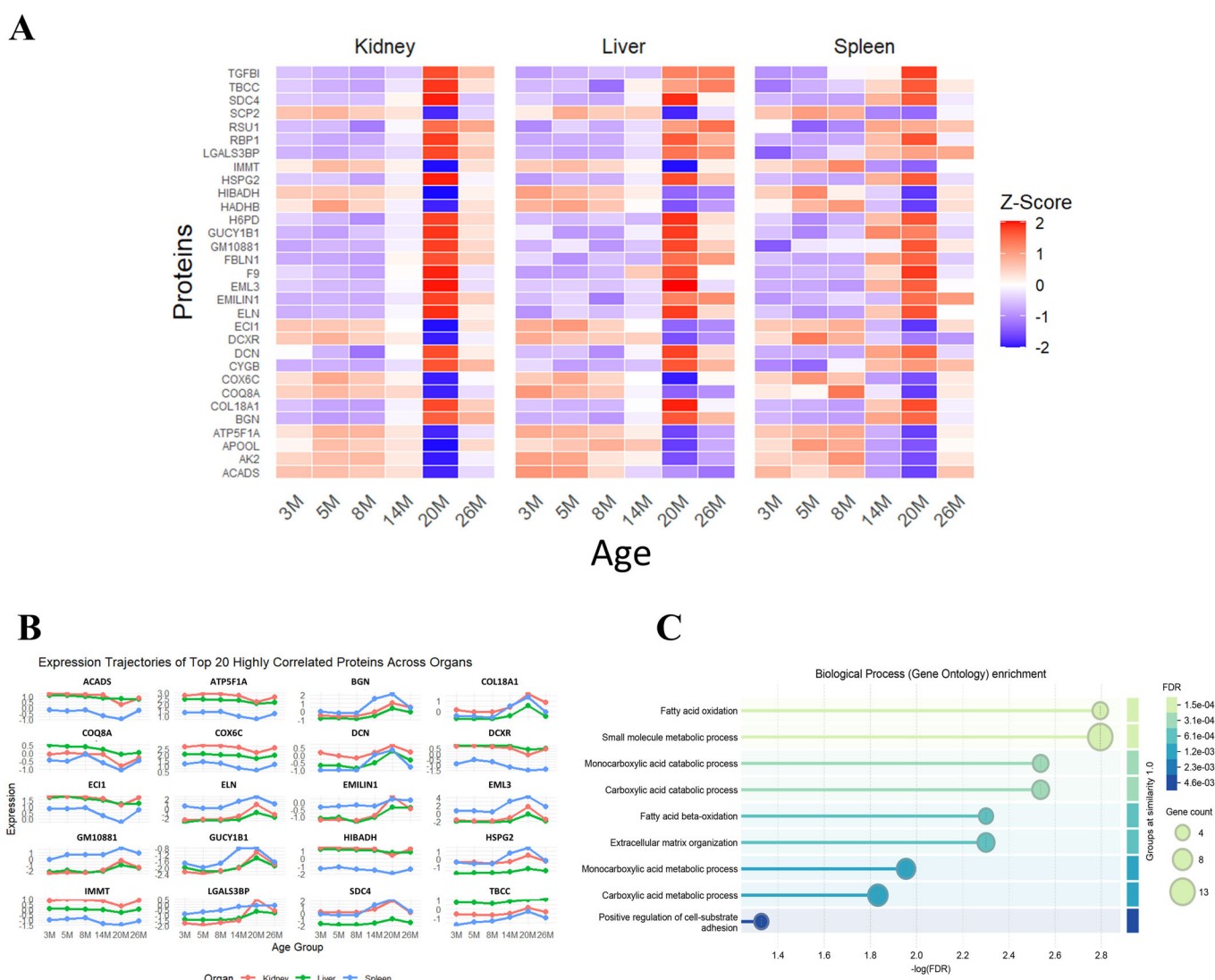

**Figure EV5. Correlation analysis of shared age-DEPs in kidney, liver and spleen.**

(A) Heatmaps showing the expression patterns of 31 highly correlated proteins (Spearman's correlation coefficient >0.5 across all pairwise organ comparisons) in kidney, liver, and spleen. The x axis represents the age groups (in months, e.g., 3 M, 5 M), while the y axis lists the proteins. The color gradient denotes normalized and log-transformed expression levels, with red indicating high expression and blue indicating low expression relative to the mean. (B) Line graphs depicting the expression trajectories of the top 20 highly correlated proteins across kidney, liver, and spleen. Each plot represents the protein expression trajectory across age groups for the three organs, providing a visual comparison of expression dynamics. The consistent trends across organs demonstrate the shared correlation of protein expression across the biological systems. (C) Gene Ontology (GO) enrichment analysis of the highly correlated proteins. The x axis represents the -log10(FDR) of enriched biological processes, while the bubble size reflects the number of proteins associated with each term. The color gradient indicates FDR significance, with lighter and darker shades denoting lower and higher significance, respectively. Source data are available on the Data Dryad public repository for this figure.

