## [Peer Review File · The EMBO Journal]

Proteomic aging signatures across mouse organs and life stages

Enzo Scifo, Sarah Morsy, Ting Liu, Kan Xie, Kristina Schaaf, Daniele Bano, and Dan Ehninger

Corresponding author(s): Dan Ehninger (Dan.Ehninger@dzne.de)

Review Timeline:

Submission Date:	11th Mar 25
Editorial Decision:	25th Mar 25
Revision Received:	17th Apr 25
Editorial Decision:	30th May 25
Revision Received:	11th Jun 25
Accepted:	16th Jun 25

Editor: Ioannis Papaioannou

Transaction Report:

Dear Dr. Ehninger,

Thank you for submitting your resource manuscript EMBOJ-2025-120525 for consideration by The EMBO Journal, and for your patience during peer review. Your manuscript has now been seen by three experts in the field, and we have received the full set of their comments, which you can find below.

As you will see, referee #1 is very supportive of your resource manuscript, pointing out that it addresses a gap in our understanding of aging proteomics with relevance to the field, while they find the study well-designed, the manuscript well-written, and the methods sufficiently detailed. Referee #2 provides a "high" rating of the novelty and significance of the report in the evaluation sheet that the reviewers are asked to return along with their comments, but he/she also identifies a few limitations in the study and provides very detailed suggestions for the improvement of the manuscript. Referee #3, on the other hand, is more critical, mentioning that he/she finds the data poorly presented and the work falling short of explaining the significance of the findings.

On balance, and considering the importance of the topic, the positive comments and recommendations of referees #1 and #2, and the fact that this is a resource manuscript rather than a primary research paper, I would like to invite you to submit a thoroughly revised version of your manuscript addressing the following referees' concerns:

- the concerns raised by referee #2 regarding statistics (major points 1 and 2, about the statistical method used to evaluate DE proteins, and about the establishment of tissue specificity of age-related changes)
- the lack of clarity on why the study used only male mice, and the particular strain, and how these could impact interpretation/generalizability of the findings (major point 3 of referee #2)
- data availability (major point 4 of referee #2); we fully agree with the referee that making not only the raw data but also processed data and scripts for their analysis available is necessary for transparent reproducible research, especially in the case of a resource manuscript; I would therefore encourage you to take the referee's advice on board and make this information available either by posting it in a permanent public repository or by using the options of Expanded View Tables and Datasets, as well as the Appendix, as part of the manuscript (please see for more information below and our guide for authors: <https://www.embopress.org/page/journal/14602075/authorguide#expandedview>)
- avoid any overstatements and improve the presentation of the data where possible, according to the comments of referee #3.

Please note that, although we agree with referee #3 that a better understanding of the significance of the observed shifts would increase the impact of your manuscript on the field, no further experimental work will be required for publication of your Resource in The EMBO Journal.

Please submit a detailed point-by-point response addressing all referees' comments along with your revised manuscript. I should add that it is The EMBO Journal policy to allow only a single round of major revision, and acceptance of your manuscript will therefore depend on the completeness of your responses in this revised version. Please let me know if you have any questions or comments that you would like to discuss with me. If there are any major points you do not agree with or cannot address during your revision, I would encourage you to share them with me as early as possible to discuss how to proceed further in the most efficient way.

While revising your manuscript, please keep in mind that The EMBO Journal has a very broad readership of molecular biologists working in many different areas of biology. I would therefore like to encourage you to revise your text as appropriate for better accessibility and clarity. The significance and relevance of the findings should be made clear -also to researchers working beyond the specific field- in the revised text, without being overstated, while alternative interpretations and limitations of the work should also be sufficiently discussed, in line also with the referees' comments and suggestions.

We generally allow three months as standard revision time (June 24, 2025). As a matter of policy, competing manuscripts published during this period will not negatively impact our assessment of the conceptual advance presented by your study. However, we request that you contact us as soon as possible upon publication of any related work, to discuss how to proceed. Should you foresee a problem in meeting this three-month deadline, please let us know in advance and we may be able to grant an extension.

Thank you for the opportunity to consider your work for publication in The EMBO Journal. I look forward to your revision.

Best regards,

Ioannis

Instructions for preparing your revised manuscript

1. When you are ready to submit the revision, please upload:

- A Word file of the manuscript text (including legends of main Figures, EV Figures and Tables). Please make sure that changes are highlighted (or "tracked") to be clearly visible.

- Individual production-quality figure files (one file per figure). When assembling your figures, please refer to our figure preparation guidelines in order to ensure proper formatting and readability in print as well as on screen:

If the data shown in a figure are obtained from n {less than or equal to} 2, please use scatter plots showing the individual data points.

i. the name of the statistical test used to generate error bars and P values

ii. the number (n) of independent experiments (please specify technical or biological replicates) underlying each data point (discussion of statistical methodology can be reported in the Materials and Methods section, but figure legends should contain a basic description of n , P , and the test applied)

iii. the nature of the bars and error bars (s.d., s.e.m.).

- A point-by-point response to the referees' comments, with a detailed description of the changes made (as a word file). All referees' concerns must be fully addressed and their suggestions taken on board. When preparing your letter of response to the referees' comments, please bear in mind that this will form part of the Review Process File and will therefore be available online to the community. Please note that you have the possibility to opt out of the transparent process at any stage prior to publication by letting the editorial office know (contact@embojournal.org); if you do opt out, the Review Process File link will point to the following statement: "No Review Process File is available with this article, as the authors have chosen not to make the review process public in this case.". For more details on our Transparent Editorial Process, please visit our website:

<https://www.embopress.org/page/journal/14602075/authorguide#transparentprocess>

- Expanded View (EV) files (replacing Supplementary Information) that are collapsible/expandable online. A maximum of 5 EV Figures can be typeset. EV Figures should be cited as "Figure EV1, Figure EV2" etc. in the text, and their respective legends should be included in the manuscript file after the legends of regular figures. See detailed instructions regarding Expanded View files here:

- For the figures that you do NOT wish to display as Expanded View figures, they should be bundled together with their legends in a single PDF file called "Appendix", which should start with a short Table of Contents (including page numbers). Appendix figures should be referred to in the main text as: "Appendix Figure S1, Appendix Figure S2" etc. Please see detailed instructions here: <https://www.embopress.org/page/journal/14602075/authorguide#expandedview>

- A complete author checklist, which you can download from our author guidelines

(<https://www.embopress.org/page/journal/14602075/authorguide>). Please note that the checklist will also be part of the Review Process File.

2. Please note that no statistics should be calculated and shown in Figures if $n=2$. Please also note that each p value should be reported as an exact value.

3. Before submitting your revision, primary datasets (and computer code, where appropriate) produced in this study need to be deposited in appropriate public databases (see <https://www.embopress.org/page/journal/14602075/authorguide#dataavailability>). The accession numbers, database, and the specific URLs (links) should be listed in a formal "Data availability" section (placed after Methods), following the example below:

"The RNA-seq datasets produced in this study are available in the following database:

Gene Expression Omnibus GSE46843 (<https://www.ncbi.nlm.nih.gov/geo/query/acc.cgi?acc=GSE46843>)"

*** All links should resolve to a page where the data can be accessed. ***

*** Please remember to provide in the Data availability section of your revised manuscript reviewer passwords if the datasets

are not yet public. ***

*** The Data Availability Section is restricted to new primary data that are part of this study. In case you have no data that require deposition in a public database, please state so instead of referring to the database: "Our study includes no data deposited in public repositories." under the heading "Data availability". ***

4. The materials and methods need to be described in the manuscript using our structured methods format, which is now required for all research articles. According to this format, the Methods section includes a single "Reagents and Tools Table" - listing key reagents, experimental models, software and relevant equipment including their sources and relevant identifiers - followed by a "Methods and Protocols" section describing the methods. Please download and fill our Reagents and Tools Table template (.docx), which you can find in our author guide:

<https://www.embopress.org/page/journal/14602075/authorguide#structuredmethods>. When submitting your revised manuscript, please do not include the Reagents and Tools Table in the Methods section of the manuscript but instead upload it as a separate file choosing the file type "Reagent Table".

5. Please check that the title and the abstract of the manuscript are brief, yet explicit, even to non-specialists. The length of the title should not exceed 100 characters, and the abstract should be a single paragraph not exceeding 175 words.

6. Please also note our reference format: <https://www.embopress.org/page/journal/14602075/authorguide#referencesformat>.

8. Please remember: digital image enhancement is acceptable practice, as long as it accurately represents the original data and conforms to community standards. If a figure has been subjected to significant electronic manipulation, this must be noted in the figure legend or in the "Materials and Methods" section. The editors reserve the right to request original versions of figures and the original images that were used to assemble the figure.

9. Our journal encourages inclusion of data citations in the reference list to directly cite datasets that were obtained from public databases. Data citations in the article text are distinct from normal bibliographical citations and should directly link to the database records from which the data can be accessed. In the main text, data citations are formatted as follows: "Data ref: Smith et al, 2001" or "Data ref: NCBI Sequence Read Archive PRJNA342805, 2017". In the Reference list, data citations must be labeled with "[DATASET]". A data reference must provide the database name, accession number/identifiers, and a resolvable link to the landing page from which the data can be accessed at the end of the reference. Further instructions are available at: <https://www.embopress.org/page/journal/14602075/authorguide#referencesformat>.

10. We request authors to consider both actual and perceived competing interests. Please review our policy (<https://www.embopress.org/page/journal/14602075/authorguide#conflictsofinterest>) and update your competing interests statement if necessary. Please name this section 'Disclosure and competing interests statement' and place it after the Acknowledgements section.

11. Please note that all corresponding authors are required to provide an ORCID ID upon submission of a revised manuscript (<https://orcid.org/>). Please find instructions on how to link your ORCID ID to your account in our manuscript tracking system in our Author guidelines (<https://www.embopress.org/page/journal/14602075/authorguide#authorshipguidelines>).

12. We use CRediT to specify the contributions of each author in the journal submission system. CRediT replaces the author contribution section, which should be removed from the manuscript. Please use the free text box to provide more detailed descriptions. See also guide to authors: <https://www.embopress.org/page/journal/14602075/authorguide#authorshipguidelines>.

14. We would also welcome the submission of cover suggestions or motifs to be used by our Graphics Illustrator in designing a cover.

15. Please use the link below to submit your revision:
<https://emboj.msubmit.net/cgi-bin/main.plex>

Referee #1:

This paper presents a well-structured and executed study that addresses a gap in our understanding of aging proteomic signatures across multiple organs and life stages in mice, with clear relevance to the aging field.

The writing is coherent and well-organized, effectively guiding the reader through the rationale, methodology, results, and conclusions. The authors have succeeded in maintaining a logical flow throughout the manuscript, which enhances the overall readability and impact of the work.

The methodology employed is robust and seems appropriately designed to answer the research question. The authors provide sufficient detail to allow for reproducibility, and the statistical analyses appear sound and well-justified.

However, I would like to note that I am not an expert in the proteomic techniques used in this study. While the methods are described clearly and seem appropriate, I encourage the editor or other reviewers with deeper expertise in this technique to ensure all technical aspects are fully validated. This suggestion is not a criticism but rather a precaution to maintain the highest standards of methodological rigor.

Referee #2:

Major Recommendation 1. The statistical method used to evaluate DE proteins is non-standard (moderated F test). Its use should be theoretically justified and validated by comparison by more standard methods. If it does not hold up under scrutiny, as I expect, the data should be reanalyzed using established statistical methods. Please compare the moderated F test results to a standard F test and to the linear trend test. It would be desirable but optional to consider a non-linear modeling technique such as generalized additive models. This is a big ask, so I am going to elaborate at length here.

The study design should determine the mode of analysis: The experimental units are individual mice, and the experimental factor of interest is age with 6 levels. The total number of mice is $n = 45$ and these are roughly evenly distributed across ages. Different mice were measured at each age, so it is not a repeated measures design. There are no other covariates or design factors to consider (although see below). The outcome is protein abundance quantified by an averaging process across several peptide abundances that are directly measured on a mass-spec. This is a nice - simple and robust - study design.

The objective is to determine if the protein changes with age. There are initially two approaches to consider.

First, we could treat age as a continuous variable (e.g., measured in days) and assume that the protein changes linearly with age. This would lead us to use a linear trend test, a linear regression of protein abundance on age. The usual test statistic is the likelihood ratio - which if we assume normal errors will follow an F distribution with $df_1 = 1$ and $df_2 = n-2$ (equivalently a t distribution with $df = n-2$). Since n is large, we could use the asymptotic chi-square ($df = 1$) distribution. If the normality assumption is in question, permutation (of age labels) can be used to obtain an empirical estimate of the p-value. The trend test is powerful and robust. This is because we make a strong assumption, linear trend, which lead to an F test with $df_1 = 1$, smaller df_1 corresponds to more power. It is robust, because even if the pattern of change with age is not linear (or errors are not normal), the trend test will still detect an overall change with age. Nonetheless, the linearity assumption is strong, and non-linearity is expected in this setting.

An alternative is to treat age as an unordered categorical variable. This has the benefit of making no assumption about how the mean abundance changes with age, but it also does not take advantage of the smooth/continuous pattern of change that is expected with age. The usual test here is a one-way ANOVA (i.e., linear regression with a single categorical variable). The likelihood ratio statistic, assuming normal errors, will follow an F distribution with $df_1 = k-1$ and $df_2 = n-k$, where k is number of age categories. Asymptotically chi-square $df = k-1$ when n is large. While this test relaxes the assumption of linearity, the space of alternatives is greatly expanded - protein abundance could zig-zag up and with age - at the cost of power. Larger df_1 corresponds to less power, and the drop in power can be dramatic even going from 1 to 2. So, the F test using categorical age is a more general but less powerful test than linear trend.

In this setting it would be reasonable to apply both tests (linear and categorical) and consider proteins that are detected as changing by either method. I would expect most DE proteins to be picked up by linear trend, and a few additional proteins where there is no overall trend, e.g. abundance goes up and then back down, to be picked up by the categorical test.

There are some intermediate approaches that allow for continuous change with age, but don't go so far as the linearity assumption. One widely used approach is the generalized additive model - which fits a smoothing 'spline' to the data. Needless to say the GAM models, despite having been in use for decades, present some choices and challenges most of which I will not get into here. The most relevant of these is how to evaluate the significance of the likelihood ratio - which will not follow any standard distribution. The R implementation of GAMs in mgcv package uses an approximation of the null distribution which is likely reliable here because the sample size is not small, the groups are roughly balanced, and the number of ages (time points) is sufficient to support a simple spline. An alternative to the approximate p-value would be to compute a permutation test by shuffling the age labels. You could use the GAM model test to detect DE proteins, but it will not be as powerful as a trend test and you will still want to do that. My recommendation would be to use the GAM model to obtain smoothed estimates of protein abundance at each age for use in displays and for clustering (across multiple proteins).

Regarding the use of t-test between adjacent time-points: I have never seen this approach used the primary method of testing, although I am sure it has been done. Pairwise comparisons among groups is generally done as a post-hoc test. Having first established that a protein is changing with age using one of the above tests, it might be of interest to know across which age

intervals changes are happening. There are other approaches, including change-point analysis, but these are likely overkill. My inclination would be to look at increments (delta) between adjacent time points from the GAM fit and treat informally, e.g., for visualization. If your goal is to determine which age increments show the greatest degree of change, you could do the pairwise testing and count significant changes.

So far we have considered evaluation of change with age for a single protein. These methods can be applied one protein at a time, with a multiple testing adjustment such a false discovery rate (please specify which software and method of FDR estimation is used!).

When testing many proteins, it may be possible to increase the power of testing by combining information across multiple proteins, e.g., using an empirical Bayes (EB) method. The moderated F test is one type of EB method. It was developed for gene expression microarray experiments, which in the early days often used very small samples sizes such as 3 vs 3 or even 2 vs 2 treatment control comparisons. Single gene test for these experiments would use t-statistics with 4 or 2 df, respectively ($F_{\{1,4\}}$ or $F_{\{1,2\}}$). The denominator of these statistics is an estimate of the within group error variance and the df (df2) indicates how much information is available (from one gene) to estimate error variance. When df2 is small, e.g. less than 10, the denominator of the t statistic can be unstable (variable) and although it will perform as expected for a single gene, the low power of this test can result in many false positive results when applied across large numbers of genes. One way to address this is to borrow information across all genes to get a better estimate of the error variance. There is an implicit assumption here that the error variances are similar - random but drawn from a common distribution. The moderated F-test replaces the denominator of the classical (one protein) F test with a 'partial pooling' EB estimate that combines data across all genes/proteins. When df2 is small this can be very effective at improving power. However, when df2 is large (>10 or certainly >20), there is little to be gained by partial pooling. Before recommending the moderated F test for this study, it would be prudent to compare the moderated and classical F test results and ask if it made any difference. One concern in this context, unlike microarrays where all genes are measured using a consistent method, the abundance estimates for proteins are based on combining a highly variable number of peptides which raises question about how similar the error variances are from protein to protein.

Major Recommendation 2: Please use an a statistical test to establish tissue-specificity of age-related changes.

The authors aim to establish that the pattern of change with age differs between tissues for a given protein. The approach used is to declare tissue specificity if change with age is significant in one tissue but not in another. There are several problems with this approach. The biggest one is possibility of type II error - when the null hypothesis is not rejected this does not establish that the protein is not changing. The correct way to establish tissue specificity is to jointly analyze the data combined across tissues. Of note here, the different tissues at any age were collected from the same mice, so we should account for that. The standard approach is a linear mixed model with a random effect term to account for within mouse correlation. Using notation from R/lme4, a 'random intercept' model can be expressed as

$$Y \sim (1 | \text{Mouse}) + \text{Age} + \text{Tissue} + \text{Age} * \text{Tissue}.$$

The term of interest is the Age*Tissue. It can be tested using a likelihood ratio test and, if significant, supports the (alternative) hypothesis of tissue specificity. The test is approximate but with a simple model like this, not a concern. Note that one can treat Age as continuous (linear trend) or categorical. We can also test all Tissues simultaneously (at some cost in power) or look at pairwise comparisons among Tissues two at a time (which might incur a multiple testing cost). Moreover, we could use GAM model in R/xx notation

$$Y \sim (1 | \text{Mouse}) + \text{Tissue} + s(\text{Age}, \text{by} = \text{Tissue})$$

which would fit a different spline for each Tissue. This model would be compared to a simpler model dropping the interaction ("by" term) to fit the same spline to both tissue and using a likelihood ratio test. Here I would default to permutation of Age labels for estimating the p-value. Of these options, the simplest, most robust and powerful will be the LMM with linear trend in Age. Some additional concerns arise because the method of obtaining tissues is not described. Were mice purchased as a single batch and tissues collected over time? Or were the mice obtained as separate cohorts and all tissues (from mice of different ages) collected at one time. Neither approach is perfect, but it would be helpful to know which was used. How were mice acclimated to their environment? How were mice housed - in groups or individually? All of these factors and more can impact the data. There may be no remedies, e.g., if cohousing is confounded with age, but it should be reported. Were the tissue samples processed in batches. These experimental details can be important, should be described, and can potentially be accounted for by including in the analysis model as covariates.

Major Recommendation 3: Please address the generality of these findings.

Please explain why the study uses only males. There are multiple publications showing that male mice are more variable than female mice is almost every measurable trait. Moreover, including both sexes does not require doubling the sample size. We have all done single sex studies, for various reasons, but it should be addressed.

Please comment on the use of C57BL/6 mice. This is also common, but it is plausible that some findings derived from the specific course of aging in this inbred strain. For reference consider the types of age-related changes reported in outbred mouse studies (e.g., PMID: 35277432 and PMID: 33687326).

Major Recommendation 4: Make the data available.

This experiment represents a valuable resource that can be reused by the research community. Providing data upon request is unreliable, at best. The raw data have been deposited in ProteomeXchange which is good and expected. However, reproducing results of this manuscript from raw data would be a daunting, likely impossible, task. The authors should post a version of intermediate processed data in publicly accessible permanent repository. Ideal this would be quantification of peptides, estimated protein abundances and accompanying analysis scripts to document the process. All in the spirit of transparent reproducible research.

Referee #3:

The study by Scifo et al. offers an organ-specific, and temporally detailed understanding of protein dynamics during aging through the use of untargeted proteomics performed in 8 mouse organs at 6 timepoints along the course of their life. It provides insights into the biological processes of aging. The analysis revealed that protein changes associated with aging occur rapidly in the kidney and spleen, moderately in the liver and lungs, minimally in skeletal muscle and testis, and subtly in the heart and brain. By identifying organ-specific pathways and protein hubs, the authors suggest this research not only highlights distinct aging mechanisms but also lays the groundwork for precision medicine strategies tailored to the unique aging processes of each organ. While I agree with the first part of the last sentence, I surely do not with the second.

The study is a descriptive exercise of the proteome changes that occur with aging. Data is very poorly presented and difficult to navigate and has relied mostly on the use of stringDB for outputs. While there is nothing wrong with such an approach, it fails to deliver a comprehensive systems approach to combine the various datasets from distinct organs.

More importantly, the study fails to deliver on a tangible understanding of what these finds actually mean. The authors find that a dramatic shift occurs at 20months of age (faster in some organs than others). The discussion is very long in an attempt to justify these findings but does not convince this reviewer of the importance of these potential findings.

More experimental work should have been done to understand and explain this shift and how and if these shifts impact aging of other organs or the overall mouse lifespan and healthspan.

Overall, I am not convinced EMBO J is an appropriate medium for the publication of such type of dataset.

Manuscript number: EMBOJ-2025-12052

Decoding Proteomic Aging Signatures Across Multiple Mouse Organs and Life Stages

We would like to thank the editor and the three reviewers for their constructive and positive suggestions. Based on their comments, we were able to improve the study and strengthen our conclusions. Below, we provide a point-by-point response to the questions raised during the review process. We hope that our revisions meet the reviewers' expectations, and the manuscript is now acceptable for publication in *EMBO J*.

Reviewers' comments and point-by-point responses

Reviewer #1.

This paper presents a well-structured and executed study that addresses a gap in our understanding of aging proteomic signatures across multiple organs and life stages in mice, with clear relevance to the aging field.

The writing is coherent and well-organized, effectively guiding the reader through the rationale, methodology, results, and conclusions. The authors have succeeded in maintaining a logical flow throughout the manuscript, which enhances the overall readability and impact of the work.

The methodology employed is robust and seems appropriately designed to answer the research question. The authors provide sufficient detail to allow for reproducibility, and the statistical analyses appear sound and well-justified.

However, I would like to note that I am not an expert in the proteomic techniques used in this study. While the methods are described clearly and seem appropriate, I encourage the editor or other reviewers with deeper expertise in this technique to ensure all technical aspects are fully validated. This suggestion is not a criticism but rather a precaution to maintain the highest standards of methodological rigor.

We thank the reviewer for their valuable feedback on our manuscript.

Reviewer #2.

Major Recommendation 1. The statistical method used to evaluate DE proteins is non-standard (moderated F test). **Its use should be theoretically justified and validated by comparison by more standard methods.** If it does not hold up under scrutiny, as I expect, the data should be reanalyzed using established statistical methods. **Please compare the moderated F test results to a standard F test and to the linear trend test. It would be desirable but optional to consider a non-linear modeling technique such as generalized additive models.** This is a big ask, so I am going to elaborate at length here.

The study design should determine the mode of analysis: The experimental units are individual mice, and the experimental factor of interest is age with 6 levels. The total number of mice is $n = 45$ and these are roughly evenly distributed across ages. Different mice were measured at each age, so it is not a repeated measures design. There are no other covariates or design factors to consider (although see below). The outcome is protein abundance quantified by an averaging process across several peptide abundances that are directly measured on a mass-spec. This is a nice - simple and robust - study design.

The objective is to determine if the protein changes with age. There are initially two approaches to consider.

First, we could treat age as a continuous variable (e.g., measured in days) and assume that the protein changes linearly with age. This would lead us to use a linear trend test, a linear

regression of protein abundance on age. The usual test statistic is the likelihood ratio - which if we assume normal errors will follow an F distribution with $df1 = 1$ and $df2 = n-2$ (equivalently a t distribution with $df = n-2$). Since n is large, we could use the asymptotic chi-square ($df = 1$) distribution. If the normality assumption is in question, permutation (of age labels) can be used to obtain an empirical estimate of the p-value. The trend test is powerful and robust. This is because we make a strong assumption, linear trend, which lead to an F test with $df1 = 1$, smaller $df1$ corresponds to more power. It is robust, because even if the pattern of change with age is not linear (or errors are not normal), the trend test will still detect an overall change with age. **Nonetheless, the linearity assumption is strong, and non-linearity is expected in this setting.**

An alternative is to treat age as an unordered categorical variable. This has the benefit of making no assumption about how the mean abundance changes with age, but it also does not take advantage of the smooth/continuous pattern of change that is expected with age. The usual test here is a one-way ANOVA (i.e., linear regression with a single categorical variable). The likelihood ratio statistic, assuming normal errors, will follow an F distribution with $df1 = k-1$ and $df2 = n-k$, where k is number of age categories. Asymptotically chi-square $df = k-1$ when n is large. While this test relaxes the assumption of linearity, the space of alternatives is greatly expanded - protein abundance could zig-zag up and with age - at the cost of power. Larger $df1$ corresponds to less power, and the drop in power can be dramatic even going from 1 to 2. So, the F test using categorical age is a more general but less powerful test than linear trend.

In this setting **it would be reasonable to apply both tests (linear and categorical) and consider proteins that are detected as changing by either method. I would expect most DE proteins to be picked up by linear trend**, and a few additional proteins where there is no overall trend, e.g. abundance goes up and then back down, to be picked up by the categorical test.

We thank the reviewer for this detailed and thoughtful comment. As suggested by the reviewer, we evaluated the performance of the moderated F-test in comparison to established statistical tests- the standard F-test and linear trend analysis, for identification of DEPs in the four organs with most protein changes (kidney, spleen, liver, and lung). This analysis indicated highly concordant results between the moderated and classical F-tests with similar number of identified DEPs, i.e. $\geq 94\%$ overlap. Moreover, in comparison to the linear trend analysis, the moderated F-test yielded slightly more DEPs for the kidney and spleen. We identified more DEPs for the liver and lung with the linear trend vs the moderated F-test; however, the extra identifications from the linear trend had marginal $R^2 \leq 0.35$ and likely represent either false positives or low confidence DEPs.

The moderated F-test (implemented via the *limma* package) uses an empirical Bayes approach to stabilize variance estimates across proteins. This is particularly helpful in proteomics datasets, where the number of peptides contributing to each protein's quantification can vary considerably, resulting in heterogeneous variance across proteins. Nonetheless, we recognize that our sample size is relatively large ($n = 45$) and degrees of freedom for residuals ($df2 \sim 39$) are high, which may reduce the need for variance moderation. Therefore, we conducted direct comparisons to assess whether the moderated test leads to substantially different results from standard methods.

Comparison to Classical F-test (Unmoderated One-way ANOVA)

We re-analyzed our data using a classical F-test with age as a categorical variable and FDR correction (Benjamini-Hochberg). The results were highly concordant with those from the moderated F-test across all organs:

Organ	% Overlap	
	Mod. F_Class. F-test	Mod. F_Linear Trend
Kidney	98.8%	86.28%
Spleen	97.2%	73.09%
Liver	95.2%	88.38%
Lung	94.4%	95.92%

We also examined the small subset of DEPs uniquely detected by each method. In all cases, these proteins had adjusted p-values tightly clustered near the FDR threshold of 0.05 and modest F-statistics. For example, in the kidney dataset, the 20 proteins unique to the moderated test had mean $F = 3.14$ and mean $\text{adj.P.Val} = 0.045$, while the 29 classical-only proteins had mean $F = 3.35$ and mean $\text{adj.P.Val} = 0.037$. This suggests that discrepancies arise from minor differences in variance estimation affecting proteins near the significance threshold, rather than systematic biases introduced by either method.

While the results of the moderated and classical F-tests are highly concordant in our dataset, the moderated approach provides a statistically principled safeguard against instability in protein-level variance estimates—especially for proteins with limited peptide coverage or noisier quantification. Its use ensures a consistent and conservative analysis framework across proteins of varying data quality, and we believe it remains appropriate for large-scale proteomics datasets such as ours.

In addition, we also performed linear trend analysis on proteins identified in the four organs with most changes (kidney, spleen, liver and lung). This analysis indicated slightly more DEPs identified for the kidney (1786 vs 1773) and spleen (1769 vs 1431) using the moderated F-test in comparison to the linear trend. In contrast, substantially less DEPs were identified in the liver (439 vs 692) and lung (147 vs 342) with the moderated F-test. Importantly, there is a substantial overlap of DEPs identified by the moderated F-test with those from the linear trend, i.e. 86%, 73%, 88% and 96% for the kidney, spleen, liver and lung. As anticipated by the reviewer, we detected more DEPs with the linear trend for the liver and lung. However, a closer evaluation indicates that the additional proteins found with only the linear trend have marginal $R^2 \leq 0.35$. As such the extra identifications obtained with the linear trend may represent either low confidence DEPs or false positives. These findings validate our approach to use the more conservative statistical approach (moderated F-test) for this study.

Conclusion

Our results demonstrate that the moderated F-test performs robustly and consistently compared to classical methods, with over 94% overlap in DEPs across organs. Differences are minor and explainable by statistical thresholding behavior near the FDR cutoff. Given the variation in peptide coverage typical in label-free proteomics data, the moderated F-test offers a stable and justifiable approach that retains consistency with classical testing strategies.

We have now added a justification for the use of moderated F-test in the introduction

Although, F-tests have not been widely used in proteomics, some recent studies (Myers, Rhoads et al., 2019, Sebastiani, Federico et al., 2021) successfully employed them. The use of the moderated F-test allows for more powerful and stable inference to detect significant changes in protein abundance compared to ordinary t-tests (Kammers, Cole et al., 2015).

In addition, we added some text on the comparison of the moderated F-test to established statistical methods in the results section.

Furthermore, we evaluated the performance of the moderated F-test in comparison to established statistical tests- the standard F-test and linear trend analysis, for identification of DEPs in the four organs with most protein changes. This analysis indicated highly concordant results between the moderated and classical F-tests with similar number of identified DEPs, i.e. $\geq 94\%$ overlap (Table EV2). The moderated F-test yielded slightly more DEPs for the kidney and spleen in comparison to the linear trend analysis. Although, we identified more DEPs for the liver and lung with the linear trend in comparison to the moderated F-test, the extra identifications from the linear trend had marginal $R^2 \leq 0.35$ (Table EV2). As such, they may represent either false positives or low confidence DEPs. Importantly, there is a substantial overlap of DEPs identified by the moderated F-test with those from the linear trend, i.e. 86%, 73%, 88% and 96% for the kidney, spleen, liver and lung (Table EV2). We therefore opted for the more conservative statistical approach (moderated F-test) that yielded robust findings, for this study.

There are some intermediate approaches that allow for continuous change with age, but don't go so far as the linearity assumption. One widely used approach is the generalized additive model - which fits a smoothing 'spline' to the data. Needless to say the GAM models, despite having been in use for decades, present some choices and challenges most of which I will not get into here. The most relevant of these is how to evaluate the significance of the likelihood ratio - which will not follow any standard distribution. The R implementation of GAMs in mgcv package uses an approximation of the null distribution which is likely reliable here because the sample size is not small, the groups are roughly balanced, and the number of ages (time points) is sufficient to support a simple spline. An alternative to the approximate p-value would be to compute a permutation test by shuffling the age labels. You could use the GAM model test to detect DE proteins, but it will not be as powerful as a trend test and you will still want to do that. **My recommendation would be to use the GAM model to obtain smoothed estimates of protein abundance at each age for use in displays and for clustering (across multiple proteins).**

Regarding the use of t-test between adjacent time-points: I have never seen this approach used the primary method of testing, although I am sure it has been done. Pairwise comparisons among groups is generally done as a post-hoc test. Having first established that a protein is changing with age using one of the above tests, it might be of interest to know across which age intervals changes are

happening. There are other approaches, including change-point analysis, but these are likely overkill. My inclination would be to look at increments (Δ) between adjacent time points from the GAM fit and treat informally, e.g., for visualization. If your goal is to determine which age increments show the greatest degree of change, you could do the pairwise testing and count significant changes. So far we have considered evaluation of change with age for a single protein. These methods can be applied one protein at a time, with a multiple testing adjustment such as a false discovery rate (please specify which software and method of FDR estimation is used!).

When testing many proteins, it may be possible to increase the power of testing by combining information across multiple proteins, e.g., using an empirical Bayes (EB) method. The moderated F test is one type of EB method. It was developed for gene expression microarray experiments, which in the early days often used very small sample sizes such as 3 vs 3 or even 2 vs 2 treatment control comparisons. Single gene test for these experiments would use t-statistics with 4 or 2 df, respectively ($F_{\{1,4\}}$ or $F_{\{1,2\}}$). The denominator of these statistics is an estimate of the within group error variance and the df (df2) indicates how much information is available (from one gene) to estimate error variance. When df2 is small, e.g. less than 10, the denominator of the t statistic can be unstable (variable) and although it will perform as expected for a single gene, the low power of this test can result in many false positive results when applied across large numbers of genes.

One way to address this is to borrow information across all genes to get a better estimate of the error variance. There is an implicit assumption here that the error variances are similar - random but drawn from a common distribution. The moderated F-test replaces the denominator of the classical (one protein) F test with a 'partial pooling' EB estimate that combines data across all genes/proteins. When df2 is small this can be very effective at improving power. However, when df2 is large (>10 or certainly >20), there is little to be gained by partial pooling. Before recommending the moderated F test for this study, it would be prudent to compare the moderated and classical F test results and ask if it made any difference. One concern in this context, unlike microarrays where all genes are measured using a consistent method, **the abundance estimates for proteins are based on combining a highly variable number of peptides which raises question about how similar the error variances are from protein to protein.**

We thank the reviewer for the detailed and helpful suggestions. Given our findings from the Standard F-test and linear trend test described above, we opted not to further explore the generalized additive model (GAM) for this study.

Major Recommendation 2: Please use an a statistical test to establish tissue-specificity of age-related changes.

The authors aim to establish that the pattern of change with age differs between tissues for a given protein. The approach used is to declare tissue specificity if change with age is significant in one tissue but not in another. There are several problems with this approach. The biggest one is possibility of type II error - when the null hypothesis is not rejected this does not establish that the protein is not changing. **The correct way to establish tissue specificity is to jointly analyze the data combined across tissues. Of note here, the different tissues at any age were collected from the same mice, so we should account for that. The standard approach is a linear mixed model with a random effect term to account for within mouse correlation.**

Using notation from R/lme4, a 'random intercept' model can be expressed as $Y \sim (1 | \text{Mouse}) + \text{Age} + \text{Tissue} + \text{Age} * \text{Tissue}$.

The term of interest is the Age*Tissue. It can be tested using a likelihood ratio test and, if significant, supports the (alternative) hypothesis of tissue specificity. The test is approximate but with a simple model like this, not a concern. Note that one can treat Age as continuous (linear trend) or

categorical. We can also test all Tissues simultaneously (at some cost in power) or look at pairwise comparisons among Tissues two at a time (which might incur a multiple testing cost). Moreover, we could use GAM model in R/xx notation $Y \sim (1 | \text{Mouse}) + \text{Tissue} + s(\text{Age}, \text{by} = \text{Tissue})$ which would fit a different spline for each Tissue.

This model would be compared to a simpler model dropping the interaction ("by" term) to fit the same spline to both tissue and using a likelihood ratio test. Here I would default to permutation of Age labels for estimating the p-value. Of these options, the simplest, most robust and powerful will be the LMM with linear trend in Age.

We thank the reviewer for this valuable comment and have now added text on tissue specificity of age-related protein changes as determined by linear mixed effects modeling, in the results section.

Organ-specific, age-related protein dynamics revealed by linear mixed-effects modeling

*To uncover tissue-specific patterns of age-related proteomic changes, we applied a linear mixed-effects model (LMM) across combined data from seven murine organs. By including a random intercept per mouse, this model accounted for intra-mouse correlations due to matched tissue sampling, thereby increasing statistical power. Specifically, the model tested for an interaction between age and tissue: $\text{Expression} \sim \text{Age} * \text{Tissue} + (1 | \text{Mouse})$. A likelihood ratio test (LRT) between the full and reduced models (excluding the interaction term) identified 2,023 proteins with significant age-by-tissue interactions (adjusted $p < 0.05$) (Table EV10), indicating divergent age trajectories across tissues.*

We further dissected these patterns by performing linear trend tests independently within each organ for each of the significant proteins, estimating both the direction and significance of age-related expression changes. This analysis enabled the construction of a comprehensive matrix summarizing directionality (Up/Down/No Change) across organs for each protein (Table EV10).

Divergent and convergent aging signatures across organs

We leveraged this matrix to categorize proteins into groups based on the consistency or divergence of their aging trajectories across organs. Across kidney, liver, and spleen, consistently upregulated proteins were enriched for immune-related processes, including immune response, cytoskeletal remodeling, and negative regulation of proteolysis, suggesting a shared immune activation signature (Table EV10). In contrast, proteins consistently downregulated across these organs showed no clear enrichment in biological processes.

Strikingly, divergent proteins between pairs of organs revealed tissue-specific aging hallmarks. For instance: proteins upregulated in liver but downregulated in spleen were enriched for mRNA splicing, RNA processing, and ribosome biogenesis. Proteins upregulated in the kidney and downregulated in the spleen were associated with fatty acid metabolism, oxidation-reduction processes, and actin filament regulation (Table EV10), potentially reflecting more robust metabolic and cytoskeletal regulatory activity in the kidney relative to the spleen. Moreover, upregulated proteins in the lung but downregulated in the kidney or liver, consistently mapped to mitochondrial function, TCA cycle, and

oxidative phosphorylation (Table EV10), implying preserved or even enhanced energy metabolism in the lung with age, contrasting with a decline in classical metabolic organs.

This comparative approach also identified a subset of proteins with organ-specific expression trends (i.e., significantly regulated with age in only one tissue). The spleen and kidney exhibited the largest number of such proteins. Functional analysis revealed that kidney-specific downregulated proteins were enriched in pathways related to fatty acid oxidation and carboxylic acid metabolism, while spleen-specific downregulated proteins mapped to translation, RNA processing, and proteostasis maintenance pathways.

These results provide a comprehensive view of how aging affects tissues in both shared and divergent ways. While certain pathways, such as immune activation, appear to be broadly upregulated with age, others, such as mitochondrial metabolism and proteostasis, show tissue-dependent regulation, potentially reflecting organ-specific resilience or vulnerability to aging stressors.

Some additional concerns arise because the method of obtaining tissues is not described. Were mice purchased as a single batch and tissues collected over time? Or were the mice obtained as separate cohorts and all tissues (from mice of different ages) collected at one time. Neither approach is perfect, but it would be helpful to know which was used. How were mice acclimated to their environment? How were mice housed - in groups or individually? All of these factors and more can impact the data. **There may be no remedies, e.g., if cohousing is confounded with age, but it should be reported. Were the tissue samples processed in batches. These experimental details can be important, should be described, and can potentially be accounted for by including in the analysis model as covariates.**

We have updated the information in the “animal housing and husbandry conditions” subsection of the Materials and methods, to address the reviewer’s concerns regarding mouse handling and tissue collection.

Animal housing and husbandry conditions

*Our study analyzed male C57BL/6J mice (stock no. 632, Jackson Laboratory), covering 6 age groups (3, 5, 8, 14, 20 and 26 months of age). All mice were purchased as one single cohort from the Jackson Laboratory. Animals were group-housed in individually ventilated cages (IVCs) under specific pathogen-free conditions (according to FELASA guidelines), with up to five animals of the same age per cage. Animal husbandry was maintained under a constant temperature of 22 °C, 55% humidity, a 12 h:12 h light/dark cycle as well as ad libitum access to food and water, in compliance with local and federal regulations regarding animal welfare. We performed daily general animal welfare monitoring and carefully checked individual animals prior to individual experiments, for their suitability according to preset criteria approved by the local animal welfare authorities specified above. Animals for which predefined humane endpoints, e.g. with ulcerating tumors, bleeding from orifice or persistent rectal prolapse were identified, were euthanized. We employed the G*Power software (v3.1.9.2) to estimate sample sizes of the mouse cohorts as required for ethical approvals of animal experiments.*

Sample preparation for mass spectrometry analysis

Animals were allowed to habituate in the dissection room for at least 30 min prior to sacrifice and tissue collection. Organs including the brain, spleen, lung, liver, kidney, testis, skeletal muscle, and heart (Fig. 1) were harvested over two consecutive days. The number of animals processed per day was balanced across age groups. Samples were snap-frozen in liquid nitrogen and stored at -80°C. Subsequently, mouse organs were pulverized in liquid nitrogen to obtain homogeneous tissue powder for protein extraction and peptide generation, followed by high-resolution accurate mass (HRAM) spectrometry. Samples were then lysed in 200 µl lysis buffer (50 mM HEPES (pH 7.4), 150 mM NaCl, 1 mM EDTA, 1.5 % SDS, 1 mM DTT; supplemented with: 1× protease and phosphatase inhibitor cocktail (ThermoScientific)). Lysis was aided by repeated cycles of sonication in a water bath (6 cycles of 1 min sonication (35 kHz) intermitted by 2 min incubation on ice). We performed silver staining to estimate total protein amounts prior to tryptic digestion. Approximately, 25 µg of total protein extracts were reduced and alkylated prior to processing by a previously described modified protocol for Filter-aided-Sample-preparation (FASP) (Scifo, Szwajda et al., 2015) to generate tryptic peptides for subsequent label-free and SureQuant-based targeted-quantitative mass spectrometry analyses. FASP is well-established in shotgun proteomics, as it facilitates the processing of various sample types at relatively low quantities while generating high-quality tryptic peptides (Wisniewski, 2016, Wisniewski, Zougman et al., 2009). Samples were digested overnight with trypsin (1:20; in 50 mM ammonium bicarbonate) directly on the filters at 37 °C and precipitated using an equal volume of 2M KCl for depletion of residual detergents. Tryptic peptides were then cleaned, desalted on C18 stage tips and re-suspended in 20 µl 1% FA for LC-MS/MS analysis. We performed MS runs with non-blinded but randomized samples and 5-9 biological replicates.

Major Recommendation 3: Please address the generality of these findings.

Please explain why the study uses only males. There are multiple publications showing that male mice are more variable than female mice is almost every measurable trait. Moreover, including both sexes does not require doubling the sample size. We have all done single sex studies, for various reasons, but it should be addressed.

We focused our analysis on male mice, as a rigorous investigation of females would require accounting for phenotypic variability linked to the estrous cycle. Moreover, our study design with 6 age groups employed 45 mice and so an additional sex would either have required an increased number of animals needed or reduced the number of animals per age group. As a compromise, we decided to use only male C57BL/6J mice for the study.

Please comment on the use of C57BL/6 mice. This is also common, but it is plausible that some findings derived from the specific course of aging in this inbred strain. For reference consider the types of age-related changes reported in outbred mouse studies (e.g., PMID: 35277432 and PMID: 33687326).

As suggested by the reviewer we have now added a justification on the choice of C57BL/6 mice for our study in the introduction section. The added part is indicated below.

We chose to carry out the present study in C57BL/6J mice, as this work is part of a larger research program investigating the effects of single-gene mutations on aging-associated proteomic changes. These mutations are maintained on a C57BL/6J genetic background, and prior studies have demonstrated lifespan extension in this strain. Moreover, C57BL/6 mice were also recently employed in recent large scale multi-organ aging studies (Schaum, Lehallier et al., 2020, Takasugi et al., 2024) thus allowing for comparative analysis with our study.

Major Recommendation 4: Make the data available.

This experiment represents a valuable resource that can be reused by the research community. Providing data upon request is unreliable, at best. The raw data have been deposited in ProteomeXchange which is good and expected. However, reproducing results of this manuscript from raw data would be a daunting, likely impossible, task. The authors should post a version of intermediate processed data in publicly accessible permanent repository. Ideal this would be quantification of peptides, estimated protein abundances and accompanying analysis scripts to document the process. All in the spirit of transparent reproducible research.

We thank the reviewer for raising this important issue and agree with their views on promoting transparent reproducible research. In addition to the raw MS data, we have also deposited peak lists (.mzML; PSI mass spectra file format; MS/MS files to verify the corresponding peptide identifications) and Search results (.mzTab and .pdResult; contain protein and PSMs for identification / quantification data) on the publicly available ProteomeXchange and JPost repository.

Raw MS data was processed in Proteome Discoverer (v3.0.1.27) and peptides exported for filtering, further processing and aggregation using QFeatures (v1.12.0). Relevant data processing steps are described in the Materials and methods. Moreover, a link to the scripts used in the analysis is also provided.

The Data availability section has been modified accordingly.

Data availability

Raw MS data, peak lists and search results are publicly available under the following accession numbers-ProteomeXchange (PXD058684) and jPOST (Okuda et al., 2017) (JPST003472) via the following link:

<https://repository.jpostdb.org/entry/JPST003472>

Analysis code is available at:

https://github.com/ehningerd/Scifo-Morsy-Liu_et_al-mouse_aging_proteome.

Reviewer #3.

The study by Scifo et al. offers an organ-specific, and temporally detailed understanding of protein dynamics during aging through the use of untargeted proteomics performed in 8 mouse organs at 6 timepoints along the course of their life. It provides insights into the biological processes of aging.

The analysis revealed that protein changes associated with aging occur rapidly in the kidney and spleen, moderately in the liver and lungs, minimally in skeletal muscle and testis, and subtly in the heart and brain. By identifying organ-specific pathways and protein hubs, the authors suggest this **research not only highlights distinct aging mechanisms but also lays the groundwork for precision medicine strategies tailored to the unique aging processes of each organ.** While I agree with the first part of the last sentence, I surely do not with the second

We thank the reviewer for her/his comments and as suggested have revised our concluding statement at the end of the introduction. The modified sentence in the manuscript now emphasizes the different magnitudes and age of onset for protein changes across organs and is indicated below.

By isolating organ-dominant pathways and protein hubs, our analysis not only highlights distinct aging mechanisms but also underscores the varied magnitude and onset of the age-related protein changes in the different organs.

The study is a descriptive exercise of the proteome changes that occur with aging. Data is very poorly presented and difficult to navigate and has relied mostly on the use of stringDB for outputs. While there is nothing wrong with such an approach, it fails to deliver a comprehensive systems approach to combine the various datasets from distinct organs.

More importantly, the study fails to deliver on a tangible understanding of what these finds actually mean. The authors find that a dramatic shift occurs at 20months of age (faster in some organs than others). The discussion is very long in an attempt to justify these findings but does not convince this reviewer of the importance of these potential findings.

More experimental work should have been done to understand and explain this shift and how and if these shifts impact aging of other organs or the overall mouse lifespan and healthspan.

Overall, I am not convinced EMBO J is an appropriate medium for the publication of such type of dataset.

We thank the reviewer for her/his comments and have attempted to improve the data presentation by incorporating the various suggestions from all the reviewers. We have for instance, made a comparison of our statistical approach (moderated F-test) to standard methods (F-test and linear trend analysis) and implemented linear mixed effects modeling to determine organ specific age-related protein dynamics, as suggested by reviewer 2 (see response to reviewer 2 above). We think that these additions have greatly improved the data presentation and hope that the modifications meet the reviewers' expectations. We agree with the reviewer that the dramatic age-related shift that occurs in the organs at 20 months merits further investigation; however, this would be a subject of another study. Overall, the comments from all the reviewers have helped us to greatly improve the manuscript and we hope that the revised manuscript addresses some of the concerns raised by the reviewer.

Dear Dr. Ehninger,

Thank you again for submitting your revised manuscript (EMBOJ-2025-120525R) to The EMBO Journal for our consideration, and for your patience during peer review. As I have already informed you, your revised manuscript has now been seen by the original referee #2, who had previously raised significant concerns in the first round of peer review. I am pleased to say that the referee now confirms that the revision has substantially improved the presentation and addressed the initially raised criticisms (the referee's comments are included below).

In light of this expert input, I am pleased to say that your manuscript has been in principle accepted for publication in The EMBO Journal. Congratulations on an excellent work.

Before we can proceed with formal acceptance and publication of the manuscript, there are a few changes and corrections we need you to make in a final version of your manuscript, before we can move forward with publication:

- During our standard plagiarism checks, we noticed that sections or even whole paragraphs of your Methods appear to have been copied from other sources (in particular, dissertations of researchers who are not listed as co-authors of the work) and used verbatim in your manuscript. We kindly request you to re-write these sections to avoid any plagiarism concerns and -if necessary- provide citations of original sources in the references list (as long as they have assigned DOIs).
- The corresponding author must be indicated on the title page of the revised manuscript.
- We noticed that funding information related to the "China Scholarship Council (CSC) (grant no. 202108080078)" is missing from our manuscript handling system (eJP); please make sure that all funding information provided in the Acknowledgements section of your manuscript is also consistently entered in eJP during resubmission.
- Please correct the reference format (the names of no more than 10 co-authors of each reference can be listed before "et al."), according to our style guidelines: <https://www.embopress.org/page/journal/14602075/authorguide#referencesformat>.
- Thank you for providing reviewer access to your deposited mass spectrometry data. This information can now be removed from the Data availability statement; please make sure that the database, ID, and specific URL of the deposited dataset will be detailed in the Data availability section of your revised manuscript, and that the data will be publicly available at the time of publication.
- The author contributions statement should be removed from the manuscript file. Instead, we use CRediT to specify the contributions of each author in the journal submission system. Please feel free to use the free text box to provide more detailed descriptions during submission. See also our guide to authors for more information: <https://www.embopress.org/page/journal/14602075/authorguide#authorshipguidelines>.
- "Materials and Methods" should be renamed to "Methods".
- All Figure panel callouts should be listed sequentially.
- We noticed that Figure callouts for Fig. 1E and 6A-I are missing.
- Figure panels are also missing in the legend of Fig. 6.
- Please make sure that your Author Checklist is uploaded to our system only once (there are currently two copies).
- Source file names, titles, legends and manuscript callouts all need to be updated to Dataset EV1-EV# instead of Tables EV1-EV4, EV6-EV15; their legends should be removed from the manuscript file and instead uploaded in a separate tab/sheet in each Excel file.
- Table EV5 should be renamed to Table EV1 with its legend included above the table in the Excel file.
- EV Figure legends should be renamed to Figure EV1-EV5 instead of Expanded View Figure 1-5.
- The Appendix file needs to be in PDF format; the header of its title page should be "Appendix for" followed by the manuscript title and a Table of Contents including page numbers of the listed items; Appendix Figures S1-S4 should be compiled in the Appendix PDF, while the Appendix Figure legends should be removed from the main manuscript file and placed instead below the corresponding Figures in the Appendix PDF file.
- Thank you for providing most of the requested Source Data. We have now checked them and it appears to us that the Source

Data files for Fig. 2A-H, 3A-E and 4H are still missing. Please provide these Data with your resubmission or clarify in your Source Data checklist.

- Please note that EMBO press papers are accompanied online by:

A) a short (2 sentences) summary of the findings and their significance,

B) 2-5 short bullet points highlighting the key results, and

C) a synopsis image in .jpg or .png format that is exactly 550 pixels wide and 300-600 pixels high (the height is variable). Please note that the text needs to be legible at the final size.

Please upload this information along with your revised manuscript (the text for A and B should be provided in a separate Word file).

- Please note that the legend for Figure 6 is not bifurcated into sub-figures in the manuscript. This needs to be rectified.

- The order of the manuscript sections must be corrected as follows: Title page - Abstract and Keywords - Introduction - Results

- Discussion - Methods - Data Availability - Acknowledgements - Disclosure and Competing Interests Statement - References - Figure Legends - main Tables (if there are any) - Expanded View Figure Legends.

Please also note that as part of the EMBO publications' Transparent Editorial Process, The EMBO Journal publishes online a Peer Review File along with each accepted manuscript. This File will be published in conjunction with your paper and will include the referee reports, your point-by-point response and all pertinent correspondence relating to the manuscript. You can opt out of this by letting the editorial office know (contact@embojournal.org). If you do opt out, the Peer Review File link will point to the following statement: "No Peer Review File is available with this article, as the authors have chosen not to make the review process public in this case."

We look forward to seeing a final version of your manuscript as soon as possible. Please let us know if you have any questions and use this link to submit your revision: <https://emboj.msubmit.net/cgi-bin/main.plex>.

Best regards,

Ioannis

Referee #2:

The authors made a substantial effort to address my concerns. On the whole, the outcomes were as expected and the revisions have substantially improved the presentation.

All editorial and formatting issues were resolved by the authors.

Dear Dan,

Congratulations on an excellent work! I am very pleased to inform you that your manuscript has been accepted for publication in The EMBO Journal. Thank you for comprehensively addressing the initially raised referees' concerns and our editorial requests for changes and corrections.

Please note that we can only process your manuscript for publication once the jSTOR dataset with ID JPST003472.3 has become publicly available, or another valid URL has been provided for this dataset. Please let us know.

Once this issue has been resolved, your manuscript will be processed for publication by EMBO Press. It will be copy edited and you will receive page proofs prior to publication. Please note that you will be contacted by Springer Nature Author Services to complete licensing and payment information.

If you have any questions, please do not hesitate to contact the Editorial Office. Thank you for your contribution to The EMBO Journal. Working with you has been a pleasure!

Best regards,

Ioannis
